# BIMODAL MASKED LANGUAGE MODELING FOR BULK RNA-SEQ AND DNA METHYLATION REPRESENTATION LEARNING

## ABSTRACT

Oncologists are increasingly relying on multiple modalities to model the complexity of diseases. Within this landscape, transcriptomic and epigenetic data have proven to be particularly instrumental and play an increasingly vital role in clinical applications. However, their integration into multimodal models remains a challenge, especially considering their high dimensionality. In this work, we present a novel bimodal model that jointly learns representations of bulk RNA-seq and DNA methylation leveraging self-supervision from masked language modeling. We leverage an architecture that reduces the memory footprint usually attributed to purely transformer-based models when dealing with long sequences. We demonstrate that the obtained bimodal embeddings can be used to fine-tune cancer-type classification and survival models that achieve state-of-the-art performance compared to unimodal models. Furthermore, we introduce a robust learning framework that maintains downstream task performance despite missing modalities, enhancing the model's applicability in real-world clinical settings.

## 1 INTRODUCTION

The growing availability of high-throughput technologies has revolutionized molecular research, generating extensive genomic, transcriptomic, and epigenomic data that holds immense potential for personalized medicine (Ho et al., 2021; Stark et al., 2019; Dai & Shen, 2022). Cancer diagnosis and prognosis thus increasingly rely on heterogeneous patient data, and the integration of these diverse data sources remains a significant challenge, even more so when some modalities may be missing in real clinical applications. The high dimensionality of each modality often makes classic machine learning and deep learning methods ineffective for diagnostic purposes. As a result, there is a growing tendency to first learn representations of the data, particularly using self-supervised approaches. In this context, foundation models have steadily emerged as powerful tools to learn effective and generalizable embeddings that can be applied to biological and clinical tasks. Trained with an unsupervised language modeling objective, they have already been applied to a wide range of omics data, including genomics (Dalla-Torre et al., 2025; Brixi et al., 2025), single-cell transcriptomics (Cui et al., 2024) or bulk RNA-seq (Gélard et al., 2025). These models extensively leverage the transformer architecture (Vaswani et al., 2017) which limits the maximum input length of the model due to the quadratic memory scaling of the attention mechanism. Recent models have developed new architectures to cope with these long-range sequences, either by integrating convolutional blocks (Avsec et al., 2021; Linder et al., 2025; Joshi et al., 2025) or state-space models (Popov et al., 2025). Among multiple studies, the Cancer Genome Atlas (TCGA, `https://portal.gdc.cancer.gov/`) is a publicly available dataset that gathers multi-omics data, in particular bulk RNA-seq and DNA methylation which are the focus of this work. Including clinical information such as survival time and divided into 33 cohorts (or cancer types), this dataset is a popular benchmark for evaluating survival analysis and cancer-type classification methods. Survival analysis, or time-to-event prediction, aims at predicting the time from diagnosis to the patient's death from the disease using censored data. Though classically tackled with Cox regression (Cox, 1972), the Cox partial likelihood has more recently been reformulated as a loss used to train deep learning architectures (Ching et al., 2018; Katzman et al., 2018).

In this paper, we introduce *MOJO*, standing for **M**ulti-**O**mics **JO**int representation learning, which we here tailor for learning joint embeddings of bulk RNA-seq and DNA methylation through bi-modal masked language modeling from the TCGA dataset. We leverage a multimodal architecture that employs a mix of convolutional and transformer layers. We show that the embeddings learned by *MOJO* lead to state-of-the-art performance in various tasks from pan-cancer cancer-type classification, survival analysis, and cancer subtype clustering. Finally, we further present a framework that allows for a downstream task model to preserve its performance in the absence of a modality by introducing an auxiliary loss based on mutual information. *The code will be made available upon acceptance*.

## 2 RELATED WORKS

**Omics representation learning**    Omics representations were usually derived from statistical methods such as Principal Component Analysis (Jolliffe, 2002) or Non-negative Matrix Factorization (NMF) (Lee & Seung, 2000), the latter being particularly suited for RNA-seq and DNA methylation due to their positivity. Deep learning architectures such as Masked Auto-Encoders (Gross et al., 2024) or Mixture-of-Experts (Meng et al., 2023) have then been applied to learn omics representations used either for survival analysis or cancer-type predictions. In line with foundation models for single-cell transcriptomics (Cui et al., 2024; Yang et al., 2022), Gélard et al. (2025) developed a transformer-based model for bulk RNA-seq representation learning. Multi-modal integration is often performed using late integration, *i.e.*, each source is encoded separately before being aggregated, often using Kronecker product (Chen et al., 2020), element-wise operations (Vale-Silva & Rohr, 2021) or cross-attention (Garau-Luis et al., 2024). On the other hand, Benkirane et al. (2023) developed a custom integration of multi-omics data using variational auto-encoders (Kingma & Welling, 2013).

**Missing modalities**    Improving the robustness of multimodal models under modality absence is crucial given the sensitivity of recent deep learning architectures to missing modalities (Ma et al., 2022). Part of the literature focuses on techniques that operate at the data level, namely leveraging modality imputation (Chen et al., 2024; Zhang et al., 2021a; Ma et al., 2021). Zhi et al. (2024) propose a retrieval-augmented in-context learning framework to address the missing modalities issue in a low-data regime. Another path towards handling missing modalities lies in adjusting the model itself with, for example, model fusion (Wagner et al., 2011) or knowledge distillation (Saha et al., 2024). Training methods are also adapted to make multimodal models robust to missing modalities by employing modality dropout (Krishna et al., 2024; Nezakati et al., 2024) during training to simulate scenarios where a subset of modalities might be missing. In our work, we adapt a technique from Ramazanova et al. (2025), which addresses the problem of missing modalities as a test-time adaptation problem, by incorporating an auxiliary loss during the fine-tuning of our model.

## 3 MULTI-OMICS JOINT REPRESENTATION LEARNING

### 3.1 BULK RNA-SEQ AND DNA METHYLATION PROCESSING FOR LANGUAGE MODELING

**Modalities alignment**    Bulk RNA-seq provides an estimate of the mean expression over all cells in a sample for a large number of genes denoted $N_{genes}$ (typically $N_{genes} \sim 10^4$). Thus, each sample of RNA-seq is composed of real values (in units of transcript per million, or TPM) per gene, $X_{rna} \in \mathbf{R}^{N_{genes}}$, to which we apply a $x \mapsto log_{10}(1 + x)$ transformation. DNA methylation is the enzymatic attachment of methyl groups to DNA's nucleotide bases (usually Cytosine followed by Guanine or *CpG* site). The methylation level of a given site is expressed by a beta value $\beta \in [0, 1]$, with the number of measured sites, $N_{sites}$ being around 450,000, resulting in a methylation sample $X_{sites\_meth} \in [0, 1]^{N_{sites}}$. The first step in our modeling is to align RNA-seq and methylation data by defining a methylation value per gene $X_{meth} \in \mathbf{R}^{N_{genes}}$. More precisely, for each gene $g \in [\![1 \; ; \; N_{genes}]\!]$, we define $sites(g)$ as all methylation sites associated with the gene, as defined in the Infinium Human Methylation 450k (Bibikova et al., 2011) BeadChip annotations. We evaluate two aggregation strategies for computing gene-level methylation values: (1) a region-weighted strategy where each site $s \in sites(g)$ is assigned weight $w(s)$ based on its genomic region, with higher weights for regulatory regions (Deaton & Bird, 2011), and (2) a simple average where all weights

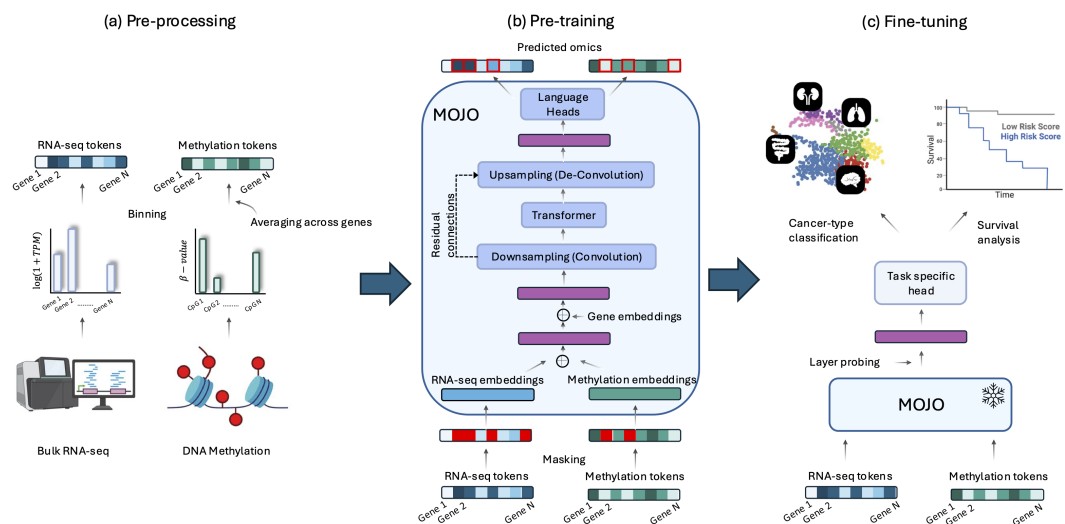

Figure 1: *MOJO* pipeline. (a) Each modality is first tokenized using linear binning. (b) *MOJO*, whose core architecture is composed of a mix of convolution and attention operations, is firstly pre-trained through bimodal masked language modeling. (c) Embeddings are probed from *MOJO* to fine-tune a task-specific head tailored for cancer-type classification or survival analysis.

are equal ($w(s) = 1$). Both compute gene-level methylation as:

$$X_{meth}[g] = \frac{\sum_{s \in sites(g)} w(s) \cdot X_{sites\_meth}[s]}{\sum_{s \in sites(g)} w(s)}$$

Benchmark experiments in the next sections of the paper demonstrate that the simple average already achieves state-of-the-art performance; thus, we use this method by default. Detailed weight specifications and a comprehensive comparison are provided in Appendix D. Finally, a bimodal sample is represented as $X = (X_{rna}, X_{meth}) \in \mathbf{R}^{2N_{genes}}$.

**Tokenization** Language models learn to estimate the likelihood of token sequences. Thus, after aligning the two modalities and obtaining a feature vector $X = (X_{rna}, X_{meth})$, each of its components is tokenized by binning their values on linear scales. The token id associated with a given RNA-seq or methylation value is its corresponding bin id, so after tokenization one sample is a vector $\widetilde{X} = (\widetilde{X}_{rna}, \widetilde{X}_{meth})$ with $\widetilde{X}_{rna} \in [\![0 \; ; \; B_{rna} - 1]\!]^{N_{genes}}$ and $\widetilde{X}_{meth} \in [\![0 \; ; \; B_{meth} - 1]\!]^{N_{genes}}$, $B_{rna}$ and $B_{meth}$ being the number of bins respectively for gene expression and methylation.

### 3.2 A LONG-RANGE MODEL ARCHITECTURE FOR BIMODAL REPRESENTATION LEARNING

In order to learn representations of bulk RNA-seq and DNA methylation, we propose a model that combines both convolution and transformer blocks. Inspired by Avsec et al. (2021); Linder et al. (2025); Joshi et al. (2025) to handle long-range genomic dependencies, this architecture allows us to cope with the high dimensionality of the two omics that we consider, each corresponding to a sequence of length $N_{genes}$. More precisely, a first bimodal embedding is obtained by passing each omic token through classic embedding layers and summing them along with a gene embedding vector. As gene expression and DNA methylation are permutation invariant, this gene embedding acts as a positional encoding and is randomly initialized. Before being fed to a transformer model made up of multi-head attention layers (Vaswani et al., 2017), the embedding is downsampled by a convolutional tower. This downscaling allows the transformer block to act on a compressed embedding vector to significantly reduce the computational cost and time. While the convolutional architecture may be counterintuitive for unordered data, it acts as an efficient mechanism for dimensionality reduction. The original sequence length is restored using a deconvolutional tower with residual connections flowing from the downsampling blocks. Separate language modeling heads predict binned gene expressions and methylation. This architecture is summarized in Figure 1 (with a zoom in on the transformer part in Appendix A.3). We also compare *MOJO* to a variant that retains only its transformer component, which we refer to as *MOJO-Transformer*.

### 3.3 Pre-training: bimodal masked language modeling

**Self-supervision loss** Our model is pre-trained through self-supervision using multimodal masked language modeling. Although this framework may be applied to more than 2 modalities that can be processed as a sequence of tokens, we present it in the bimodal case where the set of modalities $\mathbb{M} = \{rna, meth\}$. We adopt standard parameters for masked language modeling (Devlin et al., 2019): for each sequence, 15% of the tokens are corrupted to train the model. Among these corrupted tokens, 80% are replaced with a special `<MASK>` token, 10% are substituted with random tokens, and the remaining 10% are left unchanged, but still contribute to the loss. The final heads of our model provide a set of probability distributions $p_m \in [0,1]^{N_{genes} \times B_m}$ for $m \in \mathbb{M}$. Denoting $\mathcal{M}_m$, for $m \in \mathbb{M}$, the set of masked token indices for that modality, the following multimodal negative log-likelihood is then optimized:

$$\mathcal{L}_{multimodal_{MLM}} = - \sum_{m \in \mathbb{M}} \frac{1}{|\mathcal{M}_m|} \sum_{i \in \mathcal{M}_m} \log((p_m)_{i,(\tilde{X}_m)_i})$$

**Experiment** Our model is pre-trained using the TCGA dataset from 33 cohorts, resulting in 9,252 pairs of bulk RNA-seq and methylation, 5% being kept for testing. We selected $N_{genes} = 17,116$ genes by using the same set of genes as Gélard et al. (2025), from which genes with no methylation data were removed. The model is trained on a total of 192 billion tokens using the Adam (Kingma & Ba, 2014) optimizer with gradient accumulation to reach an effective batch of $3 \times 10^6$ tokens, on a TPU v4-8. The complete set of hyperparameters can be found in A.1, as well as pre-training learning curves in A.2.

## 4 Evaluation downstream tasks

The representations learned by *MOJO* are evaluated using a panel of downstream tasks ranging from supervised classification, survival analysis, and zero-shot classification, to clustering. We compare our method to unimodal (only RNA-seq or DNA methylation) and bimodal models:

**BulkRNABert** (Gélard et al., 2025): A transformer-based model, pretrained on bulk RNA-seq using masked language modeling. Embeddings are extracted from the last self-attention layer, and the mean embedding over the sequence is used as input for downstream tasks. The tokenization of the RNA-seq data is the same between *MOJO* and *BulkRNABert*, *i.e.*, the same $B_{rna}$ has been used.

**scGPT** (Cui et al., 2024): We also benchmark scGPT by adapting it from its original single-cell context to serve as a bulk RNA-seq encoder. This is achieved by applying the same fine-tuning procedure used for *BulkRNABert*, effectively repurposing scGPT for bulk data across the different tasks.

**MethFormer**: We develop the equivalent of *BulkRNABert* for DNA methylation (averaged per gene) and we will refer to it in the results as $MethFormer$. Similarly, the same value of $B_{meth}$ is maintained to allow for fair comparison. This model differs from *MethylBERT* (Jeong et al., 2025) which uses read-level methylome and not the 450k microarrays we are interested in.

**Late integration**: Bimodal integration resulting from the fusion of embeddings extracted from unimodal models. More precisely, we will refer to *Late integration (concatenation)* as the concatenation of the embeddings from *BulkRNABert* (for RNA-seq) and *MethFormer* (for methylation) which have been pre-trained beforehand. *Late integration (cross-attention)* corresponds to an integration of the two embeddings with a two-step cross-attention followed by a concatenation, allowing for interaction between the two modalities. The different cross-attention modules are only trained when fitting the downstream tasks. An illustration of the late integration is provided in Appendix B. Finally, we employ the integration strategy from **MultiSurv** Vale-Silva & Rohr (2021), where multimodal representations are aggregated by computing the maximum over features from unimodal models. We apply this aggregation method across all downstream tasks, including those beyond survival analysis, but retain the identifier *MultiSurv* to refer to this specific integration technique.

**CustOmics** (Benkirane et al., 2023): A multi-omics model based on Variational Auto-Encoders and tailored for cancer-type classification and survival analysis. Although it can handle up to three modalities (bulk RNA-seq, DNA methylation, and Copy Number Variation), we are here interested in its version that can perform the downstream tasks in the bimodal setting. Two models are con-

Table 1: Cancer type classification

| Model | test weighted-F1 |
| --- | --- |
| BulkRNABert | 0.943 ± 0.004 |
| scGPT | 0.831 ± 0.008 |
| MethFormer | 0.931 ± 0.006 |
| MOFA | 0.852 ± 0.007 |
| Late integration (concat.) | 0.945 ± 0.007 |
| Late integration (cross-att.) | 0.945 ± 0.002 |
| MultiSurv | 0.932 ± 0.005 |

| Model | test weighted-F1 |
| --- | --- |
| CustOmics (probing) | 0.911 ± 0.088 |
| CustOmics (end-to-end) | 0.946 ± 0.006 |
| MOJO - Transformer (no pre-training) | 0.827 ± 0.010 |
| MOJO - Transformer (probing) | 0.892 ± 0.009 |
| MOJO - Transformer | 0.942 ± 0.003 |
| MOJO (no pre-training) | 0.891 ± 0.006 |
| MOJO (probing) | 0.945 ± 0.006 |
| **MOJO** | **0.952 ± 0.006** |

sidered: *CustOmics (end-to-end)* that trains the VAEs and the task heads jointly, and *CustOmics (probing)* that first learns the unsupervised representation with VAEs and then uses the encoded features as input to task heads.

**Multi-Omics Factor Analysis (*MOFA*)** (Argelaguet et al., 2018): An unsupervised machine learning method designed to integrate multi-omics data by inferring a set of low-dimensional hidden factors, which can be seen as a multi-omics extension to PCA. *MOFA* factors are then used as input either to a Support Vector Machine (Cortes & Vapnik, 1995) for cancer-type classification or a Cox proportional model (Cox, 1972) for survival analysis.

We also include a more exhaustive benchmark of other feature extraction and multi-omics model integration in Appendix C.

### 4.1 CANCER-TYPE CLASSIFICATION

**Methodology** *MOJO* is first evaluated on the supervised task of cancer-type classification. The pan-cancer TCGA dataset is divided into 33 cohorts that make up the labels for the classification task. The last attention layer of the transformer component of *MOJO* is probed to obtain the embedding used for classification. After being downsampled by the convolutional layers, the embedding lies in $\mathbf{R}^{n_{\text{downsample}} \times \text{emb}_{\text{dim}}}$, with $\text{emb}_{\text{dim}} = 512$ and $n_{\text{downsample}} = 67$ (resulting from 8 successive downsampling operations by a factor of 2 from an initial sequence length of $N_{\text{genes}} = 17{,}116$, padded to the next power of 2, 17,152). Then a mean embedding in $\mathbf{R}^{\text{emb}_{\text{dim}}}$ is obtained by averaging over the sequence dimension. This embedding serves as the input to a Multi-Layer Perceptron (MLP) of two hidden layers (respectively of size 256 and 128) that outputs logits for cancer-type prediction. In addition, we add dropout (Srivastava et al., 2014) and layer normalization (Ba et al., 2016). *BulkRNABert*, *MethFormer*, and *MOJO* are further fine-tuned in addition to training the MLPs using the parameter-efficient method $IA^3$ (Liu et al., 2022), which introduces low-dimensional learnable parameters into the self-attention mechanisms and feed-forward networks. For *MOJO*, we also adapt this principle to the convolutional layers by adding a point-wise multiplication of the output of each convolutional operation with a learnable vector of the same dimension.

**Results** Cancer-type classification results on the pan-cancer TCGA dataset are presented in Table 1. For this task, the dataset has been split into 80% for training and 20% for testing (stratified by cohort), repeating the split for 5 different seeds. We report the average and standard deviation over these 5 seeds for the different metrics. We will be using both macro $F_1$ and weighted $F_1$ scores to avoid any bias due to class imbalance (label distribution is provided in Appendix C.1). *MOJO* provides state-of-the-art results when considering the two modalities, with better performance than its pure Transformer equivalent (*MOJO-Transformer*), *CustOmics* and late integration. For the latter, the *cross-attention* version performs similarly to its *concatenation* counterpart. Our joint modeling of RNA-seq and methylation with *MOJO* outperforms the corresponding unimodal transformer-based models (*BulkRNABert* and *MethFormer*). Moreover, when only probing the last attention layer and fitting an SVM (*MOJO (probing)* in the table), our model shows a clear performance increase in comparison with *CustOmics (probing)*, highlighting that representations from masked language modeling exhibit stronger predictive capacity. Finally, we show that our bimodal masked

Table 2: Average time per update step (forward + backward pass) during training of classification models on a TPU v4-8. All models are evaluated with an effective batch size of 64, achieved via gradient accumulation when necessary. For each model, we additionally report the maximum batch size supported by the model. As in classification benchmarks, parameter-efficient fine-tuning is applied to *MOJO*, *Mojo - Transformer*, and *BulkRNABert*.

| Model | Update time (seconds) | Maximum batch size |
|---|---|---|
| BulkRNABert | $4.462 \pm 0.006$ | 8 |
| Late integration (cross-attention) | $5.819 \pm 0.006$ | 4 |
| Late integration (concatenation) | $2.205 \pm 0.004$ | 16 |
| MOJO - Transformer | $17.987 \pm 0.001$ | 4 |
| MOJO | $\mathbf{0.059 \pm 0.009}$ | **1,024** |

language modeling pre-training produces a notable performance gain compared to a model trained from scratch (*MOJO (no pre-training)* in the table).

**Training times**  We additionally report in Table 2 the time required by different models (*BulkRNABert*, *Late integration (cross-attention)*, *Late integration (concatenation)*, *MOJO*, and *MOJO - Transformer*) to perform a full update step (forward and backward pass) when training a pan-cancer classification model. While supporting substantially larger batch sizes compared to purely transformer-based models or late integration mechanisms, MOJO achieves approximately a $100\times$ speedup over other benchmarked models, and a $300\times$ speedup over its transformer-based counterpart. This highlights the computational efficiency of our hybrid architecture that combines convolutional and transformer layers, offering a more scalable alternative to fully transformer-based approaches.

## 4.2 SURVIVAL ANALYSIS

**Methodology**  We then evaluate omics embeddings on a pan-cancer survival task, also known as time-to-event prediction. This task involves predicting the survival time $T_i^*$ for individuals who have cancer, specifically the time from diagnosis until death. A key challenge in survival analysis is right censoring, where the observed time $C_i^*$ might be shorter than the actual survival time $T_i^*$ due to factors like the end of a study or loss of patient contact. Consequently, the true target time used by the model, $T_i$, is defined as the minimum of the actual survival time and the censoring time ($T_i = min(T_i^*, C_i^*)$). One defines as well $\delta_i = \mathbb{1}_{T_i^* \leq C_i^*}$ (so $\delta_i = 1$ if the event occurred (death), otherwise $\delta_i = 0$), thus constituting a dataset $\mathcal{D} = \{T_i, x_i, \delta_i\}_{i=1}^N$, with $x_i$ the covariates (RNA-seq and/or methylation embeddings in our study). To ensure robust evaluation, data splits for this task were stratified based on the (cohort, event) pair, guaranteeing that each split maintains a consistent distribution of cohorts and observed mortality events. A widely used method to tackle such time-to-event problems is the Cox proportional hazards model (Cox, 1972). This semi-parametric model focuses on modeling the hazard function $\lambda(t|x)$, which represents the instantaneous rate of an event at time $t$ given covariates $x$. A Cox model expresses $\lambda$ as $\lambda(t|x) = \lambda_0(t)e^{\hat{h}_\beta(x)}$, with $\beta$ a vector of parameters (so in our case $x, \beta \in \mathbb{R}^{emb_{dim}}$), $\lambda_0(t)$ the hazard baseline, and in the Cox model, $\hat{h}_\beta(x) = \beta^T x$. More recent works (like *DeepSurv*, or *Cox-nnet*(Katzman et al., 2018; Ching et al., 2018) loosen the linear combination of features to replace $\hat{h}_\beta(x)$ with the output of a neural network, and use the negative partial Cox log-likelihood as loss:

$$\mathcal{L}_{survival} = -\sum_{i|\delta_i=1}\left(\hat{h}_\beta(x_i) - log\sum_{j\in\mathcal{R}_i}e^{\hat{h}_\beta(x_j)}\right)$$

with $\mathcal{R}_i = \{j|T_j \geq T_i\}$. For the survival task, we extract the embeddings from *BulkRNABert*, *MethFormer*, and *MOJO* in the same way as for the cancer-type classification task and train a similar MLP architecture on top. We do not consider the cross-attention version of the late integration here as the Cox loss requires working with a big enough batch size so that the computation of cross-attentions (given the sequence length of $N_{genes} = 17,116$) is not computationally efficient. Similarly, we do not apply $IA^3$ and only consider the probing experiment for the survival task.

Table 3: Pan-cancer survival analysis

| Model | C-index | Weighted C-index |
|---|---|---|
| BulkRNABert | 0.749 ± 0.003 | 0.654 ± 0.014 |
| scGPT | 0.720 ± 0.005 | 0.604 ± 0.020 |
| MethFormer | 0.736 ± 0.006 | 0.622 ± 0.014 |
| MOFA | 0.648 ± 0.037 | 0.601 ± 0.022 |
| CustOmics | 0.686 ± 0.018 | 0.639 ± 0.099 |
| Late integration (concat.) | 0.756 ± 0.004 | 0.653 ± 0.011 |
| MultiSurv | 0.742 ± 0.002 | 0.636 ± 0.011 |
| MOJO - Transformer | 0.757 ± 0.006 | 0.657 ± 0.007 |
| MOJO | **0.771 ± 0.006** | **0.670 ± 0.009** |

**Results**    Survival results on the pan-cancer TCGA dataset are presented in Table 3. The same split strategy as for the classification task is used. Two different evaluation metrics based on Harrell's C-index (Harrell et al., 1982) are reported. First, a C-index is computed on the whole test set (all cohorts) referred to as "C-index". However, in order to make sure that a pan-cancer model is able to predict survival within cohorts correctly, and not just to differentiate survival chances between cancer types, a "Weighted C-index" is also reported. This corresponds to a weighted sum of the C-indexes computed per cohort on the pan-cancer test set, with weights corresponding to the number of samples of each cohort in the test set. As for the classification task, *MOJO* exhibits higher performance than the unimodal transformer-based models and the late integration, with a significant gain over *CustOmics*. In an additional experiment, *MOJO* performance has been matched by an end-to-end version of *CustOmics* (weighted C-index of 0.669 ± 0.004). This need for end-to-end training compared to simple probing highlights the strength of *MOJO*'s learned representations. Kaplan-Meier curves are also provided in Appendix C.4, showing better patient stratification with *MOJO*.

Table 4: Zero-shot classification and clustering results on pan-cancer and PAM50 tasks. (Acc. = Accuracy, NMI = Normalized Mutual Information, ARI = Adjusted Rank Index).

| Task | Metric | MOJO | Late integration |
|---|---|---|---|
| PAM50 | Acc. | **0.777** | 0.763 |
| | NMI | **0.345** | 0.291 |
| | ARI | **0.213** | 0.154 |
| Pan-cancer | Acc. | **0.928** | 0.870 |
| | NMI | **0.862** | 0.771 |
| | ARI | **0.756** | 0.620 |

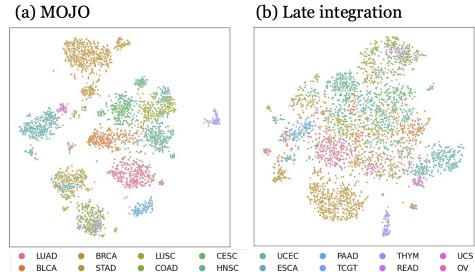

Figure 2: t-SNE representation of *MOJO* and *Late integration* embeddings, colored by cancer-type on a subset of cohorts. *MOJO*'s embeddings visually exhibit better cohort separation capacity compared to *Late Integration* ones, corroborating quantitative results from Table 4. Full pan-cancer t-SNE plot is provided in Appendix C.3.

### 4.3 ZERO-SHOT PAN-CANCER AND BREAST CANCER SUB-TYPING AND CLUSTERING

**Methodology**    To evaluate *MOJO*'s learned embeddings in a fully unsupervised manner, we assess their zero-shot classification and clustering capabilities on PAM50 breast cancer sub-typing (Luminal A, Luminal B, Basal, and HER2) (Parker et al., 2009) and the Pan-cancer dataset from section 4.1. First, zero-shot classification uses a $k$-nearest neighbors model ($k = 5$), evaluated by accuracy using 5-fold stratified cross-validation, to assess embedding quality without fine-tuning, inspired by Joshi et al. (2025). Second, Leiden clustering (Traag et al., 2019) is performed in the embedding space, with Normalized Mutual Information (NMI) and Adjusted Rand Index (ARI) as metrics. We primarily compare the effectiveness of *MOJO*'s joint modeling against late integration (concatenation) embeddings for bimodal data.

**Results**    Zero-shot classification and clustering results are shown in Table 4, showing better performance when using *MOJO* embedding than late integration. A comparison with *CustOmics* is also added in Appendix C.2, with lower performance than *MOJO*. We present in Figure 2 t-SNE (Van der Maaten & Hinton, 2008) plots of both embeddings in the pan-cancer setting, reflecting that *MOJO* embeddings more effectively separate the cohorts.

## 4.4 MOJO ENCODES KNOWN BIOLOGICAL PATHWAYS

While *MOJO*'s representations seem to be powerful for predictive tasks like cancer-type classification and survival analysis, interpreting them is also a critical step. To this end, we focus on a classification model based on *MOJO*'s embeddings (same setting as per 4.1) on the PAM50 BRCA sub-typing task (presented in section 4.3). Our classification model maps *MOJO's* embeddings of $\mathbb{R}^{\text{emb}_{\text{dim}}}$ to logits in $\mathbb{R}^{n_{classes}}$ with here $n_{classes} = 4$. We compute Shapley (SHAP) Values (Lundberg & Lee, 2017) for this model and focus on a particular subtype. We identify the most important dimension as the one that maximizes the mean absolute SHAP value. However, SHAP

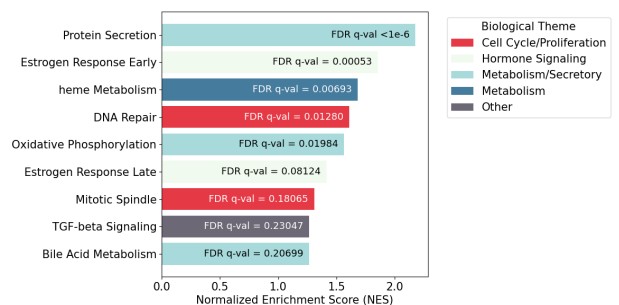

Figure 3: Cancer-relevant enriched pathways for *LuminalB* subtype using pre-rank GSEA from correlations with most informative *MOJO*'s dimensions. Only pathways with positive enrichment scores are represented. (*FDR*: False Detection Rate).

values only inform us about the contributions of the embedding dimensions to the predictions, but without any clue as to which genes have played a role in activating these dimensions. To return to the gene level, for each gene, we compute the Spearman correlation between the value of the embedding at that dimension and the bulk RNA-seq value. One ends up with a ranked list of genes that is used as input to a pre-ranked Gene Set Enrichment Analysis (GSEA, Subramanian et al. (2005)). We extend this approach to multiple dimensions by considering the average of the Fisher z-transformations of the correlation obtained for the *top k* best dimensions (as per *Stouffer*'s Z-score method (Stouffer et al., 1949)). Taking the example of *Luminal B* sub-type, Figure 3 shows that known pathways are correctly enriched (like *Protein Secretion* or *Estrogen Response*) (Li et al., 2016; Tran & Bedard, 2011; Fertig et al., 2015), showing that the dimensions that contribute the most to classifying this subtype encode known biological information ($k = 10$ has been used here). Extended results on all BRCA subtypes can be found in Appendix E.

## 4.5 VALIDATION ON EXTERNAL DATASET

To assess the generalization capability of our models beyond the TCGA training data, we performed external validation on the TARGET (Therapeutically Applicable Research to Generate Effective Treatments, https://www.cancer.gov/ccg/research/genome-sequencing/target) dataset, a cohort entirely held out from pretraining. We selected primary solid tumor samples from the GDC portal with matched bulk RNA-seq and DNA methylation (Illumina450k) data, as well as survival information, resulting in 349 patients across four cancer types (TARGET-CCSK, TARGET-NBL, TARGET-OS, and TARGET-WT). Using identical preprocessing pipelines, we evaluated model performance on the downstream tasks described above. Table 5a presents survival analysis results, while Table 5b shows zero-shot classification (discriminating between the four TARGET cohorts) and clustering performance. Detailed dataset composition and batch effect analysis between TCGA and TARGET dataset are provided in Appendix D.3.

Table 5: External validation on TARGET dataset

(a) Survival analysis

| Model | C-index | Weighted C-index |
|---|---|---|
| BulkRNABert | **0.601 ± 0.042** | 0.599 ± 0.039 |
| MethFormer | 0.548 ± 0.027 | 0.540 ± 0.042 |
| MOJO | **0.601 ± 0.023** | **0.606 ± 0.042** |

(b) Zero-shot classification and clustering

| Metric | MOJO | Late integration |
|---|---|---|
| Acc. | **0.992** | 0.991 |
| NMI | **0.949** | 0.794 |
| ARI | **0.946** | 0.706 |

## 5 MISSING MODALITIES

While *MOJO*'s joint modeling of two modalities improves downstream task performance, clinical applications may face missing modalities, with potentially one modality being entirely unavailable. Being trained by masked language modeling, *MOJO* naturally accepts missing RNA-seq or methylation information for a given subset of genes by attributing a special `<MASK>` token to these genes. One can naturally extend this procedure by attributing a sequence full of `<MASK>` tokens to a missing modality, making *MOJO* inherently capable of handling missing modalities. However, as during pre-training only a fraction of each modality is masked to account for the masked language modeling loss, the absence of a whole modality is never encountered by the model. To this end, we decided to conduct another pre-training of *MOJO* by incorporating samples from the TCGA dataset that are missing one of the two considered modalities, thus extending the initial pre-training dataset composed of 9,252 pairs $(X_{rna}, X_{meth})$ with 2,022 pairs $(X_{rna}, None)$ and 560 pairs $(None, X_{meth})$ with $None$ indicating a missing modality. We will refer to this model as *MOJO-Extended-Pretraining*. Two settings thus arise: **(S1)** one modality is either missing during model fine-tuning and we want to assess the utility of *MOJO*'s embeddings in this situation, or **(S2)** it is missing only at test-time while the downstream model has been trained with full bimodal input.

### 5.1 MISSING MODALITIES FOR FINE-TUNING (S1)

In the context of Ovarian (OV) cancer subtyping in TCGA (4 classes: differentiated, immunoreactive, mesenchymal, and proliferative (Verhaak et al., 2012)), one only gets access to RNA-seq samples. A bimodal pre-trained *MOJO* model is thus fine-tuned on this task with $(X_{rna}, None)$ as input and gets better performance than *BulkRNABert* while being faster to train (Figure 4). Using *MOJO-Extended-Pretraining* instead of *MOJO* further improves the performance in OV subtyping. In this context, we prove that *MOJO*, while initially being a bimodal model, can be used as an embedding model for RNA-seq only, and provide better performance than a specific unimodal model.

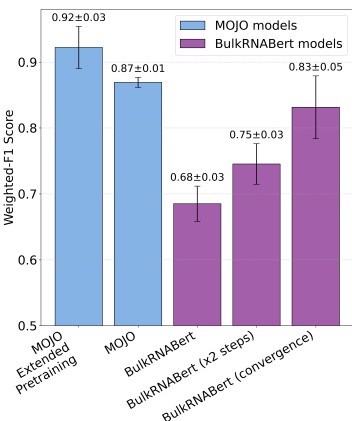

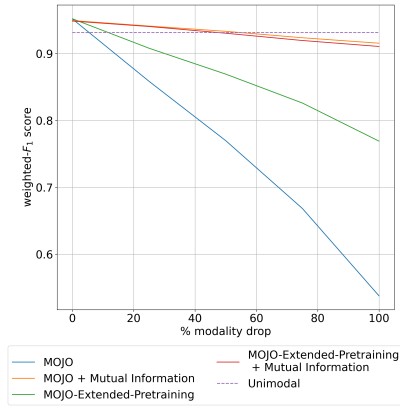

Figure 4: Ovarian cancer sub-typing: MOJO outperforms *BulkRNABert* while being faster to fine-tune. ("*BulkRNABert* models" bars from left to right: same fine-tuning budget as *MOJO*, ×2 fine-tuning steps, and until convergence). *MOJO-Extended-Pretraining* refers to an extension of the pre-training dataset with samples with missing modalities.

Figure 5: Performance when dropping RNA-seq. Test weighted-F1 score is reported as a function of the percentage of dropped RNA-seq samples in the test set. "+ *Mutual Information*" indicates the use of MI auxiliary loss during fine-tuning.

### 5.2 MISSING MODALITIES AT TEST-TIME: MUTUAL INFORMATION LOSS (S2)

**Motivation** Our goal is to create bimodal models for tasks like the pan-cancer classification task (for which both modalities are available during fine-tuning) that maintain performance comparable to unimodal models when a modality is absent at test time. Our framework begins with training a bimodal downstream model fine-tuned using complete $(X_{rna}, X_{meth})$ pairs from a pre-trained *MOJO* (per section 4.1). Our contribution involves modifying this fine-tuning for missing modalities.

**Methodology** At test time, we compute metrics under two settings: with the complete test set, and by simulating the absence of either RNA-seq or methylation by dropping it from x% of test pairs (up to x=100% for removal in all samples), all without further model fine-tuning. We only fine-tune and evaluate the downstream model with samples that have both modalities, thus allowing us to get a fair comparison between all the models. Therefore, as the dataset is fixed, one needs to change the fine-tuning procedure to cope with the drop of a full modality at test time to hope for performance maintenance. To this end, we add mutual information as an auxiliary loss paired with classic cross-entropy for the pan-cancer classification task. This quantity is used in Ramazanova et al. (2025) as a test-time adaptation technique of an audio/vision model to handle missing modalities. We adapt it to be directly incorporated during the fine-tuning phase of the model to avoid any modification of the model at test time, thus saving computational time. Following the notation from Ramazanova et al. (2025), we denote $f_\theta(x; m)$ the output of the classification model for a given input $x$ when modalities $m$ are present, with $m \in \mathcal{D}_{modality} = \{rna + meth, rna, meth\}$. One would require $f_\theta$ to provide the same prediction whatever modality is given, thus satisfying the following equality: $f_\theta(x; rna + meth) = f_\theta(x; rna) = f_\theta(x; meth)$. One can satisfy such a constraint by minimizing the following loss: $\mathcal{L}_{aux} = \mathbb{E}_{m \in \mathcal{D}_{modality}}[MI(f_\theta(x, m), m)]$, with $MI(X, Y) = D_{KL}[P_{(X,Y)} || P_X \otimes P_Y]$ corresponding to the mutual information (Shannon, 1948) between two random variables $X$ and $Y$. The mutual information is equal to 0 when the two random variables are independent; thus, minimizing this quantity as an auxiliary loss should guide the model towards providing the same output independently of the given modality. Therefore, the following loss is now optimized: $\mathcal{L} = \mathcal{L}_{task} + \lambda * \mathcal{L}_{aux}$, with $\mathcal{L}_{task}$ corresponding here to the cross-entropy for the cancer-type classification task. We detail in the algorithm of Appendix F the procedure used to compute this loss for a single example.

**Results** Results from the missing modalities experiments are shown in Figure 5 when RNA-seq is dropped (similar results when dropping DNA methylation in Appendix F). The initial *MOJO* model achieved a test weighted-$F_1$ of 0.952. Dropping all RNA-seq from the test set ($x = 100\%$) significantly decreases performance to 0.538. Adding mutual information during fine-tuning (we used $\lambda = 10$ as the ratio of the cross-entropy and the mutual information computed after model initialization) maintained stable performance (0.949) with no modality drop. However, it corrected the performance decrease when RNA-seq was missing (from 0.538 to 0.916). These recovered scores approach those of unimodal models (*i.e*, *MethFormer* as methylation is the remaining modality). Extending pre-training with pairs missing one modality (*MOJO-Extended-Pretraining*) also narrowed the performance gap without the mutual information loss, and combining *MOJO-Extended-Pretraining* with the auxiliary mutual information loss yielded performance similar to using the auxiliary loss alone.

## 6 CONCLUSION

We introduced *MOJO*, a novel methodology for learning joint representations from multi-omics data, specifically bulk RNA-seq and DNA methylation. Pre-trained using self-supervised bimodal masked language modeling, *MOJO*'s architecture, combining convolutional and attention components, efficiently handles long gene sequences, outperforming purely transformer-based unimodal models. *MOJO*'s learned embeddings achieve state-of-the-art performance in supervised tasks like cancer-typing and time-to-event prediction on the TCGA dataset compared to unimodal models. In particular, the predictive capacity of *MOJO*'s representations has been emphasized by observing a significant performance gain in the layer probing setup. Joint modeling's advantage over late integration was demonstrated through zero-shot classification and clustering. Although *MOJO* inherently accepts missing modalities, we narrowed the performance gap with unimodal models which arises when a modality is missing by incorporating a mutual information-based loss during fine-tuning. Future work includes extending *MOJO* to more data types, relaxing modality alignment requirements, extending the mutual information approach to more modalities, and improving it to exactly match unimodal performance.

## REPRODUCIBILITY STATEMENT

Towards facilitating the reproducibility of our results, we provided the pre-training setup in section 3.3 as well as both the architecture and training hyperparameters in Appendix A.1. Downstream tasks experimental settings are described in paragraph 4: data splitting, evaluation protocol, downstream task head architecture and fine-tuning procedure (probing or parameter efficient fine-tuning). *MOJO* inference code will be provided upon acceptance, as well as publicly available checkpoints on *Hugging Face*.

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

# A MOJO PRE-TRAINING

## A.1 HYPERPARAMETERS

Table 6: MOJO model and pre-training hyperparameters

| Model Hyperparameters | |
|---|---|
| Number of downsamples | 8 |
| Kernel size | 5 |
| Embedding dimension | 512 |
| Number of transformer layers | 8 |
| Feed forward dimension | 1,024 |
| Number of attention heads | 16 |
| **Training Hyperparameters** | |
| Batch size | 128 |
| Gradient accumulation | 4 |
| Learning rate | $5 \times 10^{-5}$ |
| Masking ratio | 15% |

## A.2 PRE-TRAINING LEARNING CURVES

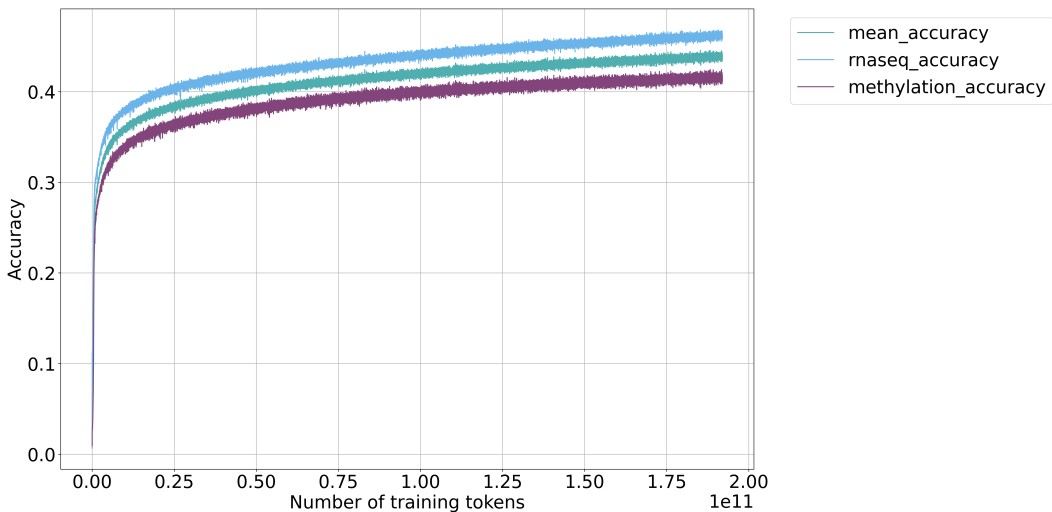

Figure 6: Bimodal masked language modeling pre-training curves of the *MOJO* architecture. The training reconstruction accuracy is represented of each omic separately as well as the average reconstruction accuracy among the different omics.

## A.3 MOJO TRANSFORMER BLOCK

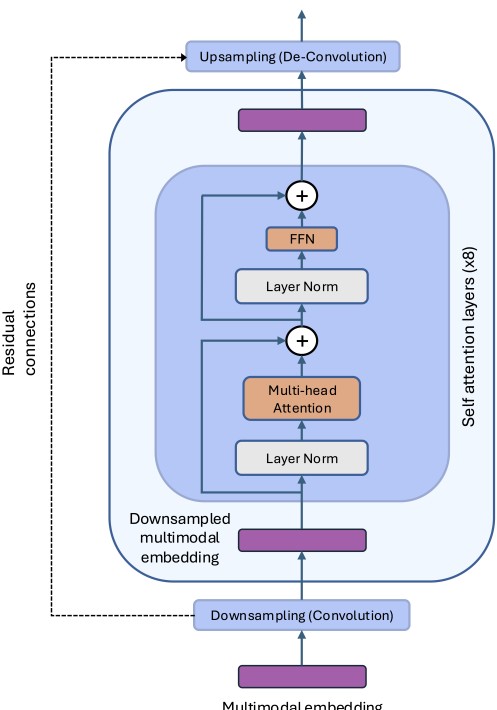

Figure 7: Zoom on the Transformer block of the *MOJO* architecture. Full architecture is provided in Figure 1.

## A.4 ARCHITECTURAL JUSTIFICATION: CONVOLUTIONS AND INPUT ORDERING

A potential concern regarding the application of convolution-based architectures, such as the U-Net, to bulk omics data is the lack of inherent spatial locality in the input. Unlike images, where pixel adjacency correlates with semantic structure, genes or CpGs in a feature vector do not possess a natural grid-like order. However, our adoption of the U-Net backbone is motivated by computational efficiency and multi-scale representation rather than spatial inductive biases. Specifically, the U-Net facilitates hierarchical dimensionality reduction, essential for handling high-dimensional inputs (hundreds of thousands of features), and enables the aggregation of signals ranging from individual gene values to broad molecular patterns.

To empirically validate that the model's performance does not rely on arbitrary input adjacency, we evaluated a variant, MOJO-ORDERED, where input features were sorted by genomic coordinates (chromosome and physical position). This ordering imposes a biological locality that a convolutional network could theoretically exploit.

Table 7: Impact of genomic ordering: performance comparison between the standard unordered MOJO and the genomically sorted MOJO-ORDERED variant. Results indicate that imposing biological order yields no statistically significant improvement, suggesting the architecture is effectively order-invariant in this context.

| | Cancer Classification | | Survival Analysis | |
|---|---|---|---|---|
| **Model** | Macro-F1 | Weighted-F1 | C-index | Weighted C-index |
| MOJO (Unordered) | $0.935 \pm 0.007$ | $0.952 \pm 0.005$ | $0.771 \pm 0.006$ | $0.670 \pm 0.009$ |
| MOJO-ORDERED | $0.936 \pm 0.006$ | $0.954 \pm 0.002$ | $0.775 \pm 0.007$ | $0.672 \pm 0.019$ |

As shown in Table 7, MOJO-ORDERED performs comparably to the standard model across both classification and survival tasks, with differences falling within the margin of error. These results confirm that the architectural benefits of MOJO stem from its capacity for hierarchical feature extraction in high-dimensional spaces, rather than local spatial correlations. Consequently, we retain the unordered design for its generality and simplicity.

### A.5 IMPACT OF TOKENIZATION GRANULARITY

We further analyzed the discretization strategy used to tokenize continuous bulk RNA-seq and DNA methylation values. Our default configuration utilizes 64 bins, aligning with prior literature (e.g., BulkRNABert (Gélard et al., 2025)) to capture sufficient nuance while mitigating the high noise inherent in raw continuous measurements. To verify the necessity of this granularity, we conducted an ablation study comparing the 64-bin scheme against a coarse-grained 5-bin alternative. The 5-bin model was pre-trained from scratch and fine-tuned on the same downstream tasks to ensure a fair comparison.

Table 8: Ablation on binning strategy. Comparison of the default 64-bin quantization versus a coarser 5-bin scheme. Finer discretization consistently outperforms the coarse-grained approach across tasks.

| Model | Cancer Classification | | Survival Analysis | |
| --- | --- | --- | --- | --- |
| | Macro-F1 | Weighted-F1 | C-index | Weighted C-index |
| MOJO (64 bins) | **0.935 ± 0.007** | **0.952 ± 0.006** | **0.771 ± 0.006** | **0.670 ± 0.009** |
| MOJO (5 bins) | 0.910 ± 0.007 | 0.937 ± 0.005 | 0.758 ± 0.009 | 0.650 ± 0.020 |

Results presented in Table 8 demonstrate that reducing the quantization resolution to 5 bins leads to a consistent degradation in performance across both tasks. For instance, the Macro-F1 score drops from $0.935$ to $0.910$, and the C-index decreases from $0.771$ to $0.758$. This suggests that while discretization acts as an effective noise regularization, preserving a moderate degree of resolution (64 bins) is crucial for capturing biologically relevant variations in expression and methylation levels.

## B LATE INTEGRATION

An illustration of the late integration mechanism is provided in Figure 8.

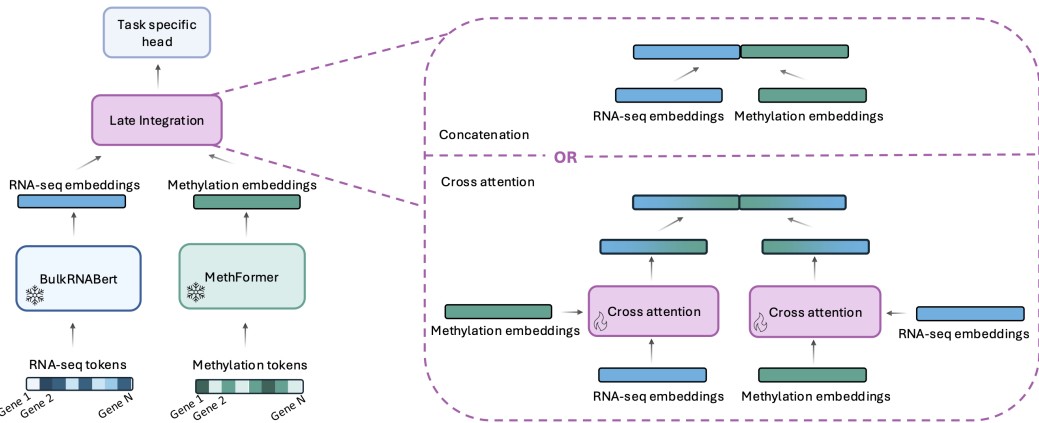

Figure 8: Late integration architecture. RNA-seq and methylation embeddings are obtained from pre-trained transformer based encoders (respectively *BulkRNABert* and *MethFormer*) and are fused either by concatenation or by a two-step cross-attention mechanism.

## C DOWNSTREAM TASKS DATASET AND BENCHMARKS

### C.1 PAN-CANCER CLASSIFICATION DATASET

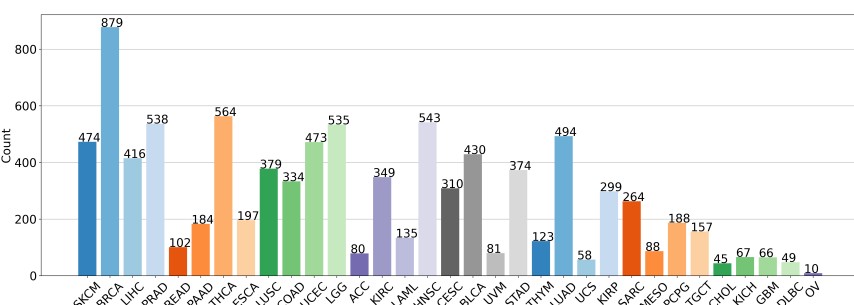

Figure 9: Pan-cancer classification label distribution.

### C.2 DOWNSTREAM TASKS BENCHMARKS

In addition to Table 1 (cancer-type classification) and Table 3 (survival analysis), a more exhaustive benchmark including other representation models for RNA-seq and DNA methylation has been performed:

- Multiple Factor Analysis (MFA) (Sánchez et al., 2012), using a latent space of dimension 256.
- Non-negative Matrix Factorization (NMF) (Lee & Seung, 2000), with the same latent space dimension as for MFA.
- *OmiEmbed* (Zhang et al., 2021b): a unified multi-task deep learning framework for multi-omics data based on Variational Auto-Encoders (Kingma & Welling, 2013) from early integrated omics.
- *IntegrAO* (Ma et al., 2024): an unsupervised framework based on Graph Neural Networks (Scarselli et al., 2008) for integrating incomplete multi-omics data, tailored for classification and survival task.

Multiple Factor Analysis and Non-negative Matrix Factorization features are then fed to a Support Vector Machine (SVM) for the cancer-type classification task and to a Cox proportional model for the survival analysis task. The results are presented in Table 9 and Table 10. We also report in Table 11 the complete benchmark on the zero-shot classification and clustering tasks which includes *CustOmics* model.

Table 9: Full benchmark on cancer-type classification

| Model | Modality | test macro-F1 | test weighted-F1 |
|---|---|---|---|
| BulkRNABert | RNA-seq | 0.918 ± 0.008 | 0.943 ± 0.004 |
| scGPT | RNA-seq | 0.793 ± 0.009 | 0.831 ± 0.008 |
| MethFormer | Methylation | 0.917 ± 0.008 | 0.931 ± 0.006 |
| MFA | Bimodal | 0.753 ± 0.013 | 0.848 ± 0.008 |
| NMF | Bimodal | 0.725 ± 0.011 | 0.827 ± 0.006 |
| MOFA | Bimodal | 0.789 ± 0.012 | 0.852 ± 0.007 |
| IntegrAO | Bimodal | 0.912 ± 0.005 | 0.911 ± 0.015 |
| OmiEmbed | Bimodal | 0.919 ± 0.004 | 0.922 ± 0.016 |
| Late integration (concatenation) | Bimodal | 0.928 ± 0.008 | 0.945 ± 0.007 |
| Late integration (cross-attention) | Bimodal | 0.929 ± 0.005 | 0.945 ± 0.002 |
| MultiSurv | Bimodal | 0.911 ± 0.005 | 0.932 ± 0.005 |
| CustOmics (probing) | Bimodal | 0.887 ± 0.065 | 0.911 ± 0.088 |
| CustOmics (end-to-end) | Bimodal | 0.922 ± 0.006 | 0.946 ± 0.006 |
| MOJO - Transformer (no pre-training) | Bimodal | 0.802 ± 0.008 | 0.827 ± 0.010 |
| MOJO - Transformer (probing) | Bimodal | 0.850 ± 0.014 | 0.892 ± 0.009 |
| MOJO - Transformer | Bimodal | 0.925 ± 0.005 | 0.942 ± 0.003 |
| MOJO (no pre-training) | Bimodal | 0.835 ± 0.015 | 0.891 ± 0.006 |
| MOJO (probing) | Bimodal | 0.928 ± 0.009 | 0.945 ± 0.006 |
| MOJO | Bimodal | **0.935 ± 0.007** | **0.952 ± 0.006** |

Table 10: Full benchmark on pan-cancer survival analysis

| Model | Modality | C-index | Weighted C-index |
|---|---|---|---|
| BulkRNABert | RNA-seq | 0.750 ± 0.004 | 0.657 ± 0.011 |
| scGPT | RNA-seq | 0.720 ± 0.005 | 0.604 ± 0.020 |
| MethFormer | Methylation | 0.735 ± 0.006 | 0.618 ± 0.017 |
| MFA | Bimodal | 0.616 ± 0.033 | 0.593 ± 0.016 |
| NMF | Bimodal | 0.616 ± 0.040 | 0.591 ± 0.025 |
| MOFA | Bimodal | 0.648 ± 0.037 | 0.601 ± 0.022 |
| IntegrAO | Bimodal | 0.710 ± 0.008 | 0.624 ± 0.006 |
| OmiEmbed | Bimodal | 0.736 ± 0.006 | 0.631 ± 0.007 |
| CustOmics | Bimodal | 0.686 ± 0.018 | 0.639 ± 0.099 |
| Late integration (concat.) | Bimodal | 0.756 ± 0.004 | 0.653 ± 0.011 |
| MultiSurv | Bimodal | 0.742 ± 0.002 | 0.636 ± 0.011 |
| MOJO - Transformer | Bimodal | 0.757 ± 0.006 | 0.670 ± 0.009 |
| MOJO | Bimodal | **0.771 ± 0.006** | **0.670 ± 0.009** |

Table 11: Full benchmark on zero-shot classification and clustering results on pan-cancer and PAM50 tasks. (Acc. = Accuracy, NMI = Normalized Mutual Information, ARI = Adjusted Rank Index).

| Task | Metric | MOJO | Late integration | CustOmics |
|---|---|---|---|---|
| | Acc. | **0.777** | 0.763 | 0.765 |
| PAM50 | NMI | **0.345** | 0.291 | 0.311 |
| | ARI | **0.213** | 0.154 | 0.176 |
| | Acc. | **0.928** | 0.870 | 0.905 |
| Pan-cancer | NMI | **0.862** | 0.771 | 0.830 |
| | ARI | **0.756** | 0.620 | 0.699 |

## C.3 BIMODAL EMBEDDINGS T-SNE VISUALISATIONS

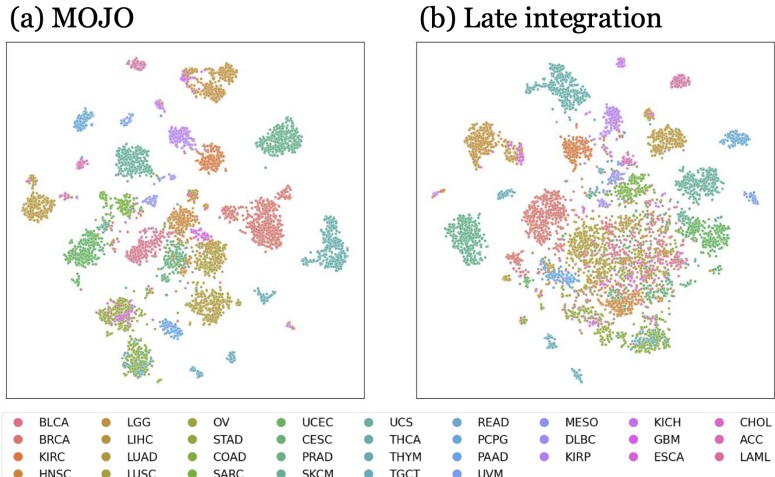

Figure 10: Pan-cancer version of the t-SNE representation of *MOJO* and *Late integration* embeddings, colored by cancer-type.

## C.4 KAPLAN-MEIER CURVES

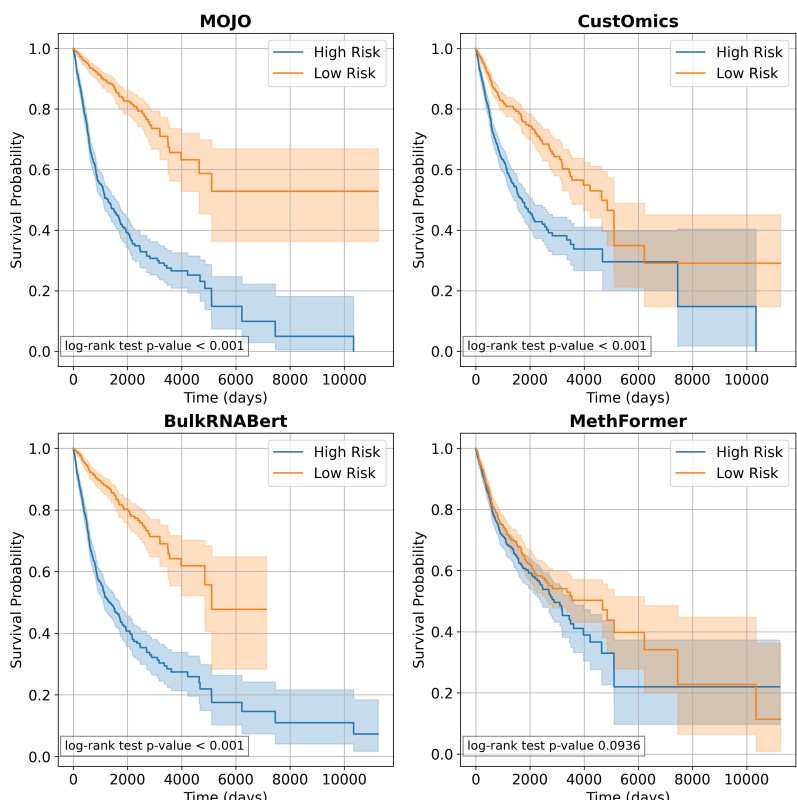

Figure 11: Kaplan-Meier curve for pan-cancer survival models for four models: *MOJO*, *CustOmics*, *BulkRNABert*, *MethFormer*. *MOJO* shows better patient stratification. "*Low Risk*" and "*High Risk*" are defined from the median of the log risk computed on the test set for each model.

## D    REGION-AWARE METHYLATION AGGREGATION

### D.1    METHODOLOGY

To map site-level methylation data to gene-level features compatible with bulk RNA-seq, in addition to the simple average of methylation values used throughout our work, we implemented a region-aware weighted aggregation strategy that accounts for the functional heterogeneity of genomic regions. We assigned differential weights to CpG sites based on their genomic location, reflecting the empirically observed distance-dependent impact of methylation on gene expression. Promoter-proximal regions (TSS200, TSS1500) received the highest weights (3.0, 2.5) due to their strong negative correlation with gene expression and established role in transcriptional silencing (Weber et al., 2007; Jones & Baylin, 2007), while 5'UTR and first exon sites received intermediate weights (2.0) based on their impact on transcriptional silencing and inhibition of elongation (Brenet et al., 2011). Gene body and 3'UTR regions served as baseline (weight = 1.0) given their weaker and often positive correlation with transcription (Ball et al., 2009), creating a biologically grounded weighting scheme that prioritizes functionally critical regulatory sites.

Additionally, we incorporated CpG island context weights to capture the differential regulatory potential of CpG sites based on their proximity to CpG islands. CpG islands (weight = 2.0) are strongly associated with gene regulation and promoter activity (Deaton & Bird, 2011), while CpG shores (weight = 1.5) have been shown to exhibit tissue-specific differential methylation with functional consequences (Irizarry et al., 2009). CpG shelves (weight = 1.0) and open sea regions (weight = 0.8) received lower weights reflecting their more variable and less consistent associations with gene expression.

The final weight for each CpG site $s$ associated with gene $g$ was computed as:

$$w(s) = w_{region}(s) \times w_{island}(s)$$

where $w_{region}(s)$ is the genomic region weight and $w_{island}(s)$ is the CpG island context weight, as detailed in Table 12.

Table 12: Weights assigned to CpG sites based on genomic region and CpG island context for region-aware methylation aggregation.

| Genomic Region | $w_{region}$ | CpG Island Context | $w_{island}$ |
|---|---|---|---|
| TSS200 | 3.0 | Island | 2.0 |
| TSS1500 | 2.5 | Shore | 1.5 |
| 5'UTR | 2.0 | Shelf | 1.0 |
| 1st Exon | 2.0 | Open Sea | 0.8 |
| Gene Body | 1.0 | | |
| 3'UTR | 1.0 | | |

The weighted methylation value for each gene was then computed as:

$$X_{meth}[g] = \frac{\sum_{s \in sites(g)} w(s) \cdot X_{sites\_meth}[s]}{\sum_{s \in sites(g)} w(s)}$$

where $sites(g)$ represents all CpG sites mapped to gene $g$ according to the Illumina HumanMethylation450 BeadChip annotations (Bibikova et al., 2011; Sandoval et al., 2011), and $X_{sites\_meth}[s]$ is the beta value for site $s$.

## D.2 ABLATION STUDY

We benchmark the weighted average of methylation value against the simple average on the pan-cancer cancer-type classification (Table 13) and survival analysis (Table 14) tasks.

Table 13: Ablation study on weighted average of methylation beta values for cancer type classification

| Non-weighted average of methylation beta value | | Weighted average of methylation beta value | |
|---|---|---|---|
| Model | Test Weighted-F1 | Model | Test Weighted-F1 |
| BulkRNABert[*] | 0.943 ± 0.004 | BulkRNABert[*] | – |
| scGPT[*] | 0.831 ± 0.008 | scGPT[*] | – |
| MethFormer | 0.931 ± 0.006 | MethFormer | 0.935 ± 0.005 |
| MFA | 0.848 ± 0.008 | MFA | 0.850 ± 0.009 |
| NMF | 0.827 ± 0.006 | NMF | 0.824 ± 0.006 |
| MOFA | 0.852 ± 0.007 | MOFA | 0.849 ± 0.008 |
| IntegrAO | 0.911 ± 0.015 | IntegrAO | 0.944 ± 0.015 |
| OmiEmbed | 0.922 ± 0.016 | OmiEmbed | 0.925 ± 0.017 |
| Late integration (concat.) | 0.945 ± 0.007 | Late int. (concat.) | 0.940 ± 0.005 |
| Late integration (cross-att.) | 0.945 ± 0.002 | Late int. (cross-att.) | 0.941 ± 0.004 |
| MultiSurv | 0.911 ± 0.005 | Late int. (max.) | 0.930 ± 0.010 |
| CustOmics (probing) | 0.911 ± 0.088 | CustOmics (probing) | 0.893 ± 0.071 |
| CustOmics (e2e[**]) | 0.946 ± 0.006 | CustOmics (e2e[**]) | 0.941 ± 0.003 |
| MOJO-Transformer (probing) | 0.892 ± 0.009 | MOJO-Transformer (probing) | 0.896 ± 0.007 |
| MOJO-Transformer | 0.942 ± 0.003 | MOJO-Transformer | 0.938 ± 0.010 |
| MOJO (probing) | 0.945 ± 0.006 | MOJO (probing) | 0.950 ± 0.006 |
| MOJO | **0.952 ± 0.006** | MOJO | 0.951 ± 0.005 |

[*]BulkRNABert and scGPT use only RNA data and are unaffected by methylation preprocessing. [**]e2e = end-to-end

Table 14: Ablation study on weighted average of methylation beta values for survival analysis

| Non-weighted average of methylation beta value | | Weighted average of methylation beta value | |
|---|---|---|---|
| Model | Weighted C-index | Model | Weighted C-index |
| BulkRNABert[*] | 0.657 ± 0.011 | BulkRNABert[*] | – |
| scGPT[*] | 0.604 ± 0.020 | scGPT[*] | – |
| MethFormer | 0.618 ± 0.017 | MethFormer | 0.622 ± 0.018 |
| MFA | 0.593 ± 0.016 | MFA | 0.594 ± 0.018 |
| NMF | 0.591 ± 0.025 | NMF | 0.590 ± 0.026 |
| MOFA | 0.601 ± 0.022 | MOFA | 0.598 ± 0.024 |
| IntegrAO | 0.624 ± 0.006 | IntegrAO | 0.625 ± 0.006 |
| OmiEmbed | 0.631 ± 0.007 | OmiEmbed | 0.633 ± 0.007 |
| CustOmics | 0.639 ± 0.099 | CustOmics | 0.642 ± 0.023 |
| Late integration (concat.) | 0.653 ± 0.011 | Late integration (concat.) | 0.652 ± 0.016 |
| MultiSurv | 0.636 ± 0.011 | MultiSurv | 0.635 ± 0.021 |
| MOJO-Transformer | 0.657 ± 0.007 | MOJO-Transformer | 0.641 ± 0.010 |
| MOJO | **0.670 ± 0.009** | MOJO | 0.654 ± 0.018 |

[*]BulkRNABert and scGPT use only RNA data and are unaffected by methylation preprocessing.

While the region-aware weighted methylation aggregation strategy was designed to better capture the differential regulatory potential of CpG sites, our empirical evaluation reveals nuanced results. As shown in Tables 13 and 14, this biologically motivated weighting scheme does not yield improvements for the MOJO model in either cancer-type classification or survival prediction tasks; therefore, we retain the simpler average-based aggregation in our final model. This confirms that *MOJO* is able

to learn the relevant regulatory structure directly from the data without requiring hand-crafted methylation priors. However, when examining the *MethFormer* baseline, we observe marginal improvements with region-aware aggregation. Furthermore, during pre-training, the region-aware scheme achieves consistently higher reconstruction accuracy compared to simple averaging (Figure 12). This improved reconstruction performance suggests that weighted aggregation, by accounting for the differential regulatory roles of promoters, gene bodies, and CpG islands, yields a more biologically grounded representation of methylation patterns. Nevertheless, these improvements do not transfer to the multimodal setting, indicating that the additional biological signal captured by region-aware weighting may be redundant with information already encoded within the RNA-seq modality. This finding highlights a broader challenge in multimodal learning: biologically motivated features that improve unimodal representation quality do not necessarily enhance multimodal fusion when modalities contain overlapping biological signals.

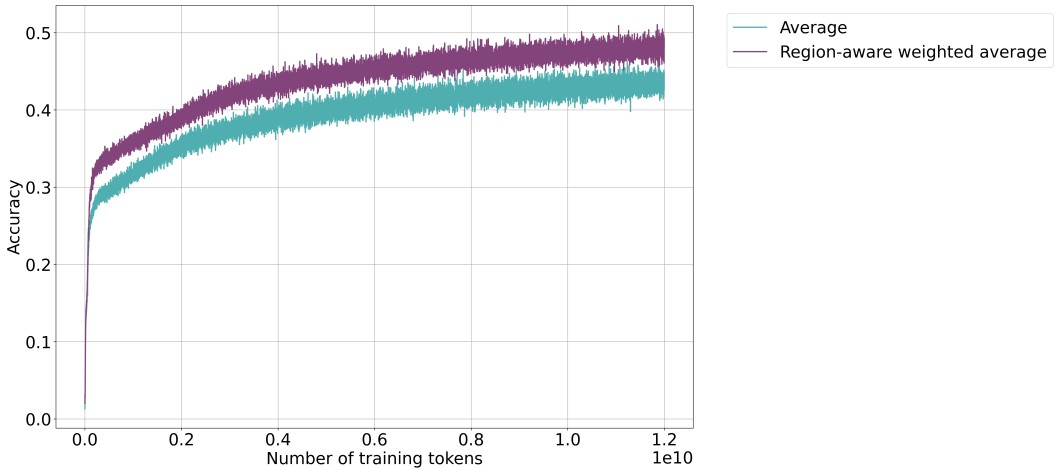

Figure 12: Comparison of *MethFormer* pre-training learning curves for the different methylation preprocessing schemes.

## D.3 TARGET DATASET

### D.3.1 DATASET COMPOSITION

Table 15: Composition of the TARGET dataset

| Cohort | Cancer type | # samples | Events | Censored | Event rate |
|---|---|---|---|---|---|
| TARGET-CCSK | Clear Cell Sarcoma of the Kidney | 11 | 3 | 8 | 27.3% |
| TARGET-NBL | Neuroblastoma | 133 | 66 | 67 | 49.6% |
| TARGET-OS | Osteosarcoma | 83 | 29 | 54 | 34.9% |
| TARGET-WT | Wilms Tumor (Kidney) | 122 | 50 | 72 | 41.0% |
| **Total** | **All Cohorts** | **349** | **148** | **201** | **42.4%** |

### D.3.2 BATCH EFFECT ANALYSIS

We quantified cross-platform batch effects using three complementary metrics: (1) Silhouette scores to measure global dataset separation (range: -1 to 1, with 0 indicating complete mixing), (2) the k-nearest neighbor Batch Effect Test (kBET) to assess whether batch labels are randomly distributed in local neighborhoods (range: 0 to 1, with 1 indicating perfect mixing), and (3) the Pearson correlation of mean bulk RNA-seq and DNA methylation levels between platforms. Our analysis revealed weak batch effects (bulk RNA-seq: Silhouette=0.19, kBET=0.95, r=0.86; DNA Methylation: Silhouette=0.12, kBET=0.92, r=0.96). Additionally, PCA visualization showed substantial overlap between TCGA and TARGET samples (Appendix D.3, Figure 13). The high kBET scores and strong

correlations demonstrate excellent local mixing and overall concordance, validating the model's cross-platform generalization capabilities.

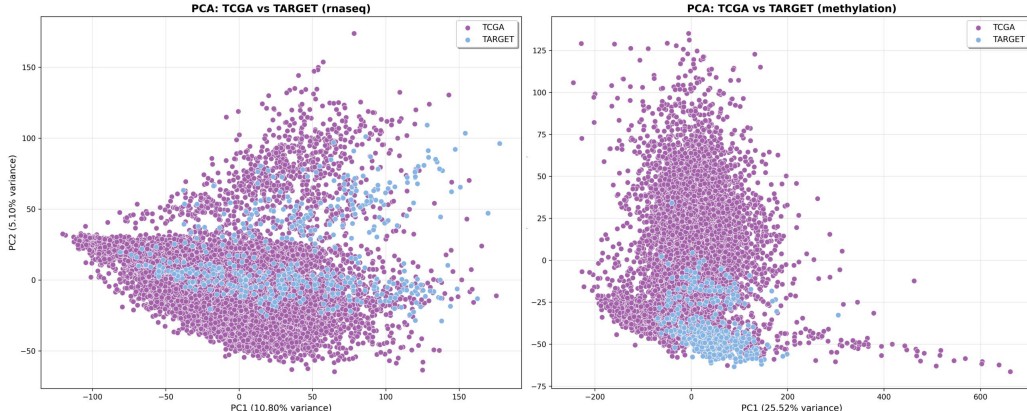

Figure 13: Bulk RNA-seq and DNA Methylation PCA plots - comparison of TCGA and TARGET datasets.

# E    INTERPRETABILITY RESULTS

In this section, we provide additional results for the interpretability section 4.4. Figure 14 provides a swarm plot of the SHAP values for the *Luminal B* subtype of BRCA. It demonstrates a strong monotonic impact of the most influential dimensions on the prediction of the *Luminal B* subtype. For a given dimension, its high values are associated with SHAP values of a consistent sign, while its low values are associated with the opposite, indicating that the model has learned distinct predictive patterns. Figure 15 provides the pre-rank GSEA results for all the four BRCA subtypes. We only kept the enriched pathways with false detection rate (FDR) q-value less than 0.25 (Subramanian et al., 2005) and we present the pathways that have a positive enrichment score.

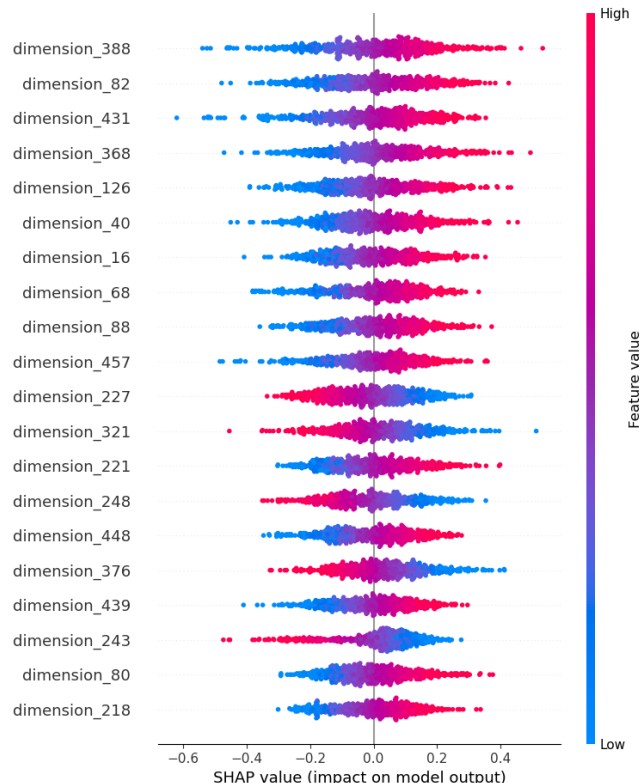

Figure 14: Shapley Values swarmplot for *LumB* sub-type of BRCA cancer.

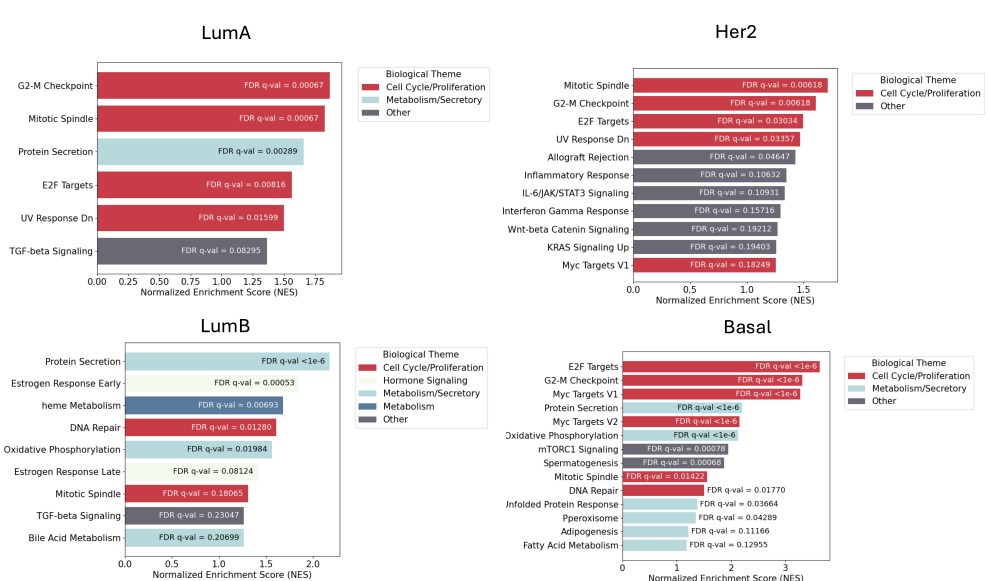

Figure 15: Pre-rank gene set enrichment analysis results for all subtypes of BRCA from bulk RNA-seq correlations with most predictive *MOJO*'s embedding dimensions.

# F    MISSING MODALITIES EXPERIMENTS

Algorithm 1 details the procedure to compute the mutual information loss for a single example. Figure 16 provides the complete results from the missing modalities experiments (when either RNA-seq or methylation is missing), while Table 16 provides exact figures.

---

**Algorithm 1** Mutual information auxiliary (MI) loss

---

**Input:** Omics tokens $X = \{rnaseq : x_{rnaseq}, meth : x_{meth}\}$, true class label $y$, sequence length $N$, mask token `<MASK>`, mutual information coefficient $\lambda$, classification model $f_\theta$
**Output:** single example loss
**if** $noMissingModality(X)$ **then**
$\quad modalities = [rna + meth, rnaseq, meth]$
$\quad output = [f_\theta(X)]$
$\quad$**for** $m \in [rnaseq, meth]$ **do**
$\quad\quad X' \leftarrow copy(X)$
$\quad\quad X'[m] \leftarrow [\text{<MASK>}] * N$
$\quad\quad output.append(f_\theta(X'))$
$\quad$**end for**
$\quad MILoss = MI(output, modalities)$
**else**
$\quad MILoss = 0.0$
**end if**
Loss $= CrossEntropy(f_\theta(X), y) + \lambda * MILoss$

---

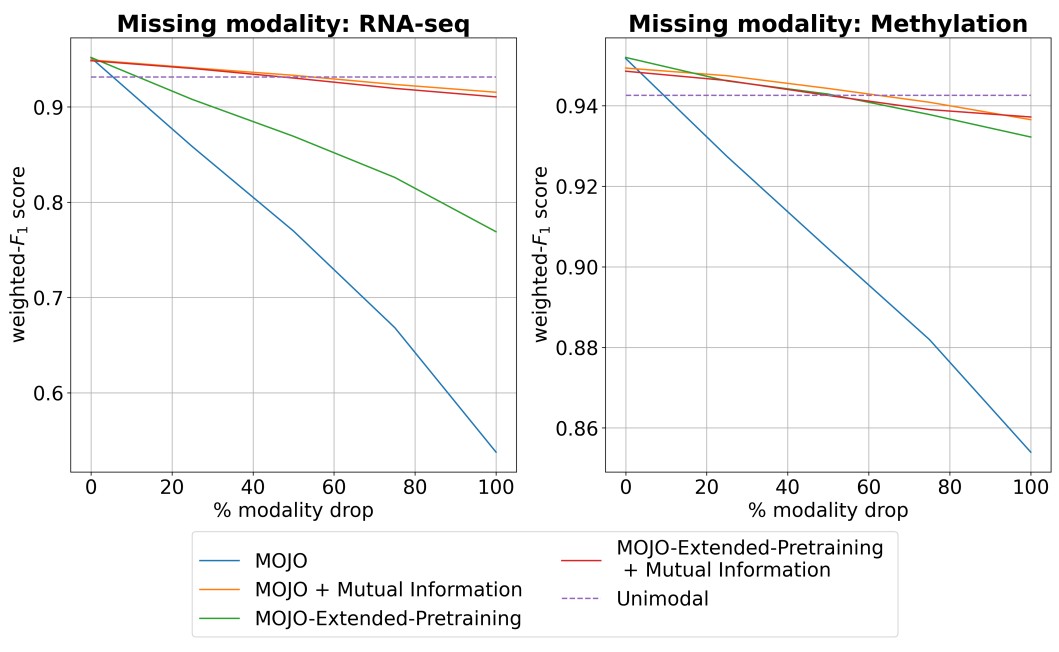

Figure 16: Missing modalities experimental results. Test weighted-$F_1$ score for the pan-cancer classification is reported for different methods to handle the absence of a modality in x% of the samples (left: RNA-seq, right: Methylation). Unimodal models are respectively *MethFormer* and *BulkRNABert* when RNA-seq or Methylation is missing.

Table 16: Missing modalities experiment: cancer type classification

| Model | Add mutual information | Drop modality (test time) | macro-F1 | test weighted-F1 |
|---|---|---|---|---|
| BulkRNABert | ✗ | - | 0.918 ± 0.008 | 0.943 ± 0.004 |
| MethFormer | ✗ | - | 0.917 ± 0.008 | 0.931 ± 0.006 |
| MOJO | ✗ | - | 0.935 ± 0.007 | 0.952 ± 0.006 |
| MOJO | ✗ | Drop 100% of RNA-seq | 0.422 ± 0.022 | 0.538 ± 0.025 |
| MOJO | ✗ | Drop 100% of Methylation | 0.764 ± 0.024 | 0.854 ± 0.011 |
| MOJO | ✓ | - | 0.930 ± 0.007 | 0.949 ± 0.004 |
| MOJO | ✓ | Drop 100% of RNA-seq | 0.895 ± 0.008 | 0.916 ± 0.007 |
| MOJO | ✓ | Drop 100% of Methylation | 0.911 ± 0.012 | 0.937 ± 0.008 |
| MOJO-Extended-Pretraining | ✗ | - | 0.933 ± 0.006 | 0.952 ± 0.003 |
| MOJO-Extended-Pretraining | ✗ | Drop 100% of RNA-seq | 0.653 ± 0.013 | 0.769 ± 0.004 |
| MOJO-Extended-Pretraining | ✗ | Drop 100% of Methylation | 0.903 ± 0.010 | 0.932 ± 0.005 |
| MOJO-Extended-Pretraining | ✓ | - | 0.929 ± 0.006 | 0.949 ± 0.005 |
| MOJO-Extended-Pretraining | ✓ | Drop 100% of RNA-seq | 0.883 ± 0.005 | 0.911 ± 0.004 |
| MOJO-Extended-Pretraining | ✓ | Drop 100% of Methylation | 0.911 ± 0.010 | 0.937 ± 0.006 |

# G  USE OF LARGE LANGUAGE MODELS

We acknowledge the use of *Gemini 2.5 Pro* as an assistive tool for minor grammatical corrections and stylistic refinements to improve readability. All conceptual work and written content are the original contribution of the authors.

