# OpenReview forum: "Bimodal masked language modeling for bulk RNA-seq and DNA methylation representation learning"
_ICLR.cc/2026/Conference — Submitted to ICLR 2026_

### Official Review · Reviewer_Rh7r · 2025-10-29

**Soundness:** 2
**Presentation:** 4
**Contribution:** 3
**Rating:** 6
**Confidence:** 4

**Summary:**

MOJO: Multi-Omics JOint representation learning is a novel approach to jointly model bulk RNA-seq and DNA methylation data. It does early fusion of both data modalities and learns a mixed representation via transformers blocks in a learned lower dimensional space, providing efficient training and inference for high-dimensional genomics data. The authors demonstrate strong performance of MOJO in downstream applications and modeling settings. They further interpret their model in the context of biological pathways and provide an interesting analysis on approaches to address missing data modalities.

**Strengths:**

- The authors present an efficient mechanism to do multihead attention in a lower dimensional space, demonstrating impressive gains in model training efficiency (e.g. Table 2). This is especially relevant for genomics data, as in the current setting, where the number of features is very large and can further scale if incorporating further genomic data types.

- Strong results across tasks for cancer classification, survival analysis, and zero-shot subtyping. Particularly interesting analysis with section 4.4 analysing the alignment of latent model features with known biological pathways.

- Nice study on missing data modalities that have strong implications for further experimentation in research or clinical settings. Both approaches for extended pretraining with missing data modalities using entire-modality masked tokens and the mutual information auxiliary loss are interesting and demonstrate strong results in this biomedical setting.

**Weaknesses:**

- For downstream objectives (Table 1 and Table 3), while the pretraining and downstream (classification or survival) objectives do differ, there is still potential data leakage of using some subset of the pretraining data in the test splits of the cross-validated classification results. This is a major flaw of the paper particularly when other RNA-seq and methylation data is present in the GDC alongside TCGA.

- Table 1: late-attention with a newly trained cross-attention module is comparable to probed MOJO. While the authors claim PEFT of pretrained MOJO does best, no comparable performance to a PEFT of pretrained BulkRNABert+MethFormer with a cross attention module is done (unless the illustration of the cross-attention approach in appendix B Figure 7 is incorrect and the modality specific encoders are tuned along with the cross attention?). If, in late integration, a PEFT of the modality arms and training the cross-attention match the PEFT of MOJO, then the results are further tempered.

- The analysis of embedding correlation with biological pathways is very interesting, however the author’s claim that pathways are “correctly enriched” on line 380 needs an appropriate reference.

**Questions:**

- Would be interesting to see an ablation where the convolutional dimensionality reduction is not used

- On line 149-151, why convolutions instead of fully connected layers for downsampling?

- Line 251 the authors say the results do not significantly differ - is this backed by a statistical test or is this loose use of the word? should be clarified

- Are data splits (for pretraining or downstream cross validation) stratified by cancer type and observed mortality?

- Is late integration in section 4.3 also restricted to concatenation? Should be explicitly stated

- Further clarification of the KNN method in section 4.3 should be provided (w.r.t if results are cross validated, what data was used for training, etc)

---

> ### Author Response · Authors · 2025-11-26
> **Response to reviewer Rh7r (1)**
>
> Dear Reviewer,
>
> We thank you once again for your review. In addition to the general response addressed to all reviewers to consolidate common feedback, we provide below detailed responses to your specific comments:
>
> 1. >For downstream objectives (Table 1 and Table 3), while the pretraining and downstream (classification or survival) objectives do differ, there is still potential data leakage of using some subset of the pretraining data in the test splits of the cross-validated classification results. This is a major flaw of the paper particularly when other RNA-seq and methylation data is present in the GDC alongside TCGA.
>
> We thank the reviewer for raising this important concern regarding potential data leakage. We acknowledge that utilizing the TCGA cohort for both pre-training and downstream evaluation constitutes a limitation, even though the pre-training objective (masked language modeling) differs fundamentally from the downstream tasks (classification and survival prediction). While self-supervised pre-training learns general representations without access to the labels or clinical outcomes used downstream, we recognize that strictly separating pre-training and evaluation sets provides a methodologically stronger evaluation. To address this limitation and rigorously demonstrate generalization, we conducted an external validation using the TARGET dataset, which was entirely held out from the pre-training phase. This analysis is detailed in the **"External validation on an independent cohort"** section of our general response. Importantly, the results from this completely independent cohort demonstrate that our model generalizes effectively to unseen data.
>
> 2. >Table 1: late-attention with a newly trained cross-attention module is comparable to probed MOJO. While the authors claim PEFT of pretrained MOJO does best, no comparable performance to a PEFT of pretrained BulkRNABert+MethFormer with a cross attention module is done (unless the illustration of the cross-attention approach in appendix B Figure 7 is incorrect and the modality specific encoders are tuned along with the cross attention?). If, in late integration, a PEFT of the modality arms and training the cross-attention match the PEFT of MOJO, then the results are further tempered.
>
> We thank the reviewer for this thoughtful observation and the opportunity to clarify the design rationale behind our late integration baseline. We would like to maintain that our late integration approaches rely on frozen pretrained encoders, with training restricted to the fusion module (in this instance, the cross-attention mechanism). Introducing parameter-efficient fine-tuning (PEFT) to the modality-specific encoders would alter this paradigm, shifting it from a strict "late integration" strategy to a "mixed integration" approach that is conceptually closer to MOJO's design. Furthermore, we note a practical constraint: applying PEFT to both the BulkRNABert and MethFormer full Transformer architectures would incur substantially higher computational costs than our other benchmarked methods, including MOJO, rendering the comparison less equitable in terms of computational budget.
>
>
> 3. >The analysis of embedding correlation with biological pathways is very interesting, however the author’s claim that pathways are “correctly enriched” on line 380 needs an appropriate reference
>
> We appreciate the reviewer’s suggestion to provide appropriate references supporting our claim regarding pathway enrichment. We have now added citations to substantiate that Luminal B tumors are estrogen-receptor–positive and display active estrogen-regulated transcriptional programs, consistent with the strong enrichment of the Estrogen Response Early pathway observed in our analysis [1]. Furthermore, we have included references confirming that Luminal-type breast cancers exhibit elevated expression of secreted factors and paracrine-signaling components, supporting the enrichment of protein-secretion–related pathways [2].
>
> References:
>
> [1] Tran, B., & Bedard, P. L. (2011). Luminal-B breast cancer and novel therapeutic targets. Breast Cancer Research, 13(6), 221.
>
> [2] Fertig, E. J., Lee, E., Pandey, N. B., & Popel, A. S. (2015). Analysis of gene expression of secreted factors associated with breast cancer metastases in breast cancer subtypes. Scientific Reports, 5, 12133.

---

> > ### Author Response · Authors · 2025-11-26
> > **Response to reviewer Rh7r (2)**
> >
> > 4. >Would be interesting to see an ablation where the convolutional dimensionality reduction is not used
> >
> > We thank the reviewer for suggesting an ablation study regarding the use of convolutional dimensionality reduction in the MOJO architecture. We have provided a detailed unified response in the general comment addressed to all reviewers (under the paragraph **“MOJO architecture benchmark”**), as several reviews requested benchmarks on specific components of the model. We highlight in that analysis that our hybrid architecture yields superior performance compared to its pure Transformer counterpart while remaining computationally more efficient.
> >
> >
> > 5. >On line 149-151, why convolutions instead of fully connected layers for downsampling?
> >
> > We appreciate the reviewer’s inquiry regarding our choice of convolutional layers over fully connected layers for downsampling. Our primary motivation is computational and parameter efficiency: given an initial sequence length of 17,116, a single fully connected layer would require over 146 million parameters just for the first downsampling step (17,116 × 8,558), whereas convolutional layers achieve the equivalent 8× downsampling ($2^8 = 256\times$ total reduction) with orders of magnitude fewer parameters. Moreover, directly compressing the sequence to its final length in a single step would result in significant information loss. Consequently, we adopted this stepwise reduction strategy, which is consistent with typical findings in work utilizing U-Net architectures.
> >
> > 6. >Line 251 the authors say the results do not significantly differ - is this backed by a statistical test or is this loose use of the word? should be clarified
> >
> > We thank the reviewer for seeking clarification on our terminology. At line 251, we compare the performance of the two late integration schemas on the pancancer classification task, where "concatenation" yielded a macro F1-score of $0.928 \pm 0.008$ and "cross-attention" yielded $0.929 \pm 0.005$. Our statement regarding the lack of significant difference was intended to highlight that the confidence intervals overlap, with the "cross-attention" mean exceeding "concatenation" by only 0.001, rather than claiming the result of a formal statistical hypothesis test.
> >
> >
> > 7. >Are data splits (for pre-training or downstream cross validation) stratified by cancer type and observed mortality?
> >
> > We thank the reviewer for the opportunity to clarify our data splitting strategy. We confirm that stratification was applied exclusively to the downstream tasks and not during pre-training. For the cancer-type classification task, dataset splits were stratified to ensure equal representation of each cohort across the different splits. For the survival analysis task, splits were stratified based on the (cohort, event) pair to guarantee that each split maintains a consistent distribution of cohorts and observed mortality events.
> >
> > 8. >Is late integration in section 4.3 also restricted to concatenation? Should be explicitly stated
> >
> > We thank the reviewer for requesting this clarification. Section 4.3 evaluates the capacity of learned embeddings to perform zero-shot classification for pancancer typing and breast cancer sub-typing, comparing MOJO against a Late Integration baseline. In this specific context, the Late Integration approach was restricted to concatenation. We did not employ the "cross-attention" variant because it necessitates training the fusion modules, which is incompatible with a true zero-shot prediction setting. We are grateful to the reviewer for noting that this was not explicitly stated, and we have updated the manuscript to include this explanation.
> >
> >
> > 9. >Further clarification of the KNN method in section 4.3 should be provided (w.r.t if results are cross validated, what data was used for training, etc)
> >
> >
> > We appreciate the opportunity to clarify the K-Nearest Neighbors (KNN) methodology used in Section 4.3. For this zero-shot prediction analysis, we utilized the full TCGA dataset for the pan-cancer task and the BRCA cohort subset for the PAM50 task. KNN accuracies were computed using 5-fold stratified cross-validation, with stratification applied to ensure balanced representation of the target labels (cancer cohort for the pan-cancer task and BRCA subtype for the PAM50 task) within each fold. We acknowledge that the use of cross-validation was not explicitly detailed in the original manuscript, and we have updated the text to clearly reflect this procedure.

---

> > > ### Comment · Reviewer_Rh7r · 2025-11-26
> > >
> > > Thank you for addressing the concerns and questions in our review.   I think our scores are a good fit for the paper.

---

### Official Review · Reviewer_GBM6 · 2025-10-31

**Soundness:** 2
**Presentation:** 3
**Contribution:** 2
**Rating:** 4
**Confidence:** 2

**Summary:**

The paper introduces MOJO, a bimodal masked-language modeling framework for bulk RNA-seq and DNA methylation. Both modalities are aligned at the gene level and discretized into tokens (linear binning). A conv-downsampling + Transformer backbone is pre-trained with bimodal MLM and then probed/fine-tuned for pan-cancer type classification, survival analysis, and some zero-shot subtyping/clustering. To handle missing modalities at test time, the authors add an auxiliary mutual-information–based consistency loss during fine-tuning. The paper reports strong in-domain results and notable training efficiency.

**Strengths:**

- Clear, easy-to-reproduce pipeline: gene-level alignment + simple tokenization; architecture and objectives are standard and well explained.

- Training efficiency: the hybrid conv+Transformer design scales to long gene sequences with large batch sizes and fast steps.

- Solid in-domain performance: competitive results on TCGA for classification and survival; additional zero-shot analyses are included.

- Missing-modality robustness: the MI-based consistency objective substantially improves performance when one modality is absent.

- Initial interpretability: SHAP/GSEA-style analyses provide some biological intuition for the learned representations.

**Weaknesses:**

1. Evaluation loop / external generalization is limited. Pretraining and downstream evaluation are both conducted within TCGA splits, without independent validation on ICGC/GEO or cross-platform cohorts.

2. Coarse cross-modal alignment. Aggregating methylation to per-gene values via “near-gene CpG averaging” risks discarding promoter/enhancer/genic-region specificity and may mask true regulatory dependencies.

3. Modest novelty. The approach largely combines known ingredients (linear binning → MLM pretraining; conv downsampling + Transformer; standard probing/finetuning). Contributions feel incremental rather than conceptually new.

**Questions:**

1. External generalization. Beyond TCGA, can you replicate key results on independent cohorts (e.g., ICGC/GEO) and report cross-platform robustness (batch effects, sequencing platforms)? Please include both classification and survival, plus the zero-shot analyses, with identical preprocessing and fixed splits.

2. Alignment strategy. How sensitive are results to the per-gene methylation aggregation? Could you compare against region-aware or annotation-aware schemes (e.g., promoter/CGI/shore weighting, enhancer links, or graph-based aggregation), and provide a quantitative ablation (including different binning schemes or continuous embeddings without discretization)?

---

> ### Author Response · Authors · 2025-11-26
> **Response to reviewer GBM6**
>
> Dear Reviewer,
>
> We thank you once again for your insightful review. We appreciate that your two primary concerns align with questions raised by other reviewers, underscoring their significance. To ensure a thorough and consistent response, we have addressed these points in dedicated sections of our general response. Below, we specify which paragraphs correspond to each of your specific comments.
>
> 1. >External generalization. Beyond TCGA, can you replicate key results on independent cohorts (e.g., ICGC/GEO) and report cross-platform robustness (batch effects, sequencing platforms)? Please include both classification and survival, plus the zero-shot analyses, with identical preprocessing and fixed splits.
>
> We thank the reviewer for emphasizing the necessity of external validation to demonstrate generalization beyond the TCGA dataset. Since this important point was raised by multiple reviewers, we have provided a detailed analysis in the **“External validation on an independent cohort”** section of our general response. As demonstrated therein, our validation on the independent TARGET dataset confirms that MOJO maintains high performance across classification and survival tasks, exhibiting strong robustness to out-of-domain generalization.
>
> 2. >Alignment strategy. How sensitive are results to the per-gene methylation aggregation? Could you compare against region-aware or annotation-aware schemes (e.g., promoter/CGI/shore weighting, enhancer links, or graph-based aggregation), and provide a quantitative ablation (including different binning schemes or continuous embeddings without discretization)?
>
> We appreciate the reviewer's close attention to our alignment strategy. We have addressed the sensitivity of our results to per-gene methylation aggregation in two complementary sections of our general response. The **"Methylation 450k data preprocessing"** section details our comparison between simple averaging and a more complex region-aware weighted aggregation scheme, while the **"MOJO architecture benchmark"** section provides quantitative ablation studies on tokenization and binning strategies. These analyses reveal that region-aware aggregation yields comparable downstream performance to simple averaging, supporting our decision to retain the simpler design. Additionally, regarding binning strategies, our results demonstrate that our initial configuration significantly outperforms the alternative strategy utilizing a reduced number of bins.

---

### Official Review · Reviewer_Yhbd · 2025-10-31

**Soundness:** 2
**Presentation:** 3
**Contribution:** 3
**Rating:** 4
**Confidence:** 4

**Summary:**

This paper proposes MOJO, a bimodal model for learning joint representations from bulk RNA-seq and DNA methylation data. It utilizes a self-supervised bimodal masked language modeling (MLM) task for pre-training. The architecture combines a U-Net-like convolutional structure for efficient dimensionality reduction with a Transformer core to integrate the two modalities. Evaluated on the TCGA dataset, MOJO shows strong performance on downstream tasks like cancer-type classification and survival analysis, outperforming single-modality models.

**Strengths:**

1. This work pioneers the integration of bulk RNA-seq and DNA methylation, uniting complementary transcriptomic and epigenetic data for deeper mechanistic insight.

2. Its focus on clinically accessible bulk omics significantly enhances translational value, bridging foundational research with practical implementation.

**Weaknesses:**

1. An oversimplified architectural diagram (Fig. 1) that abstracts the core Transformer structure.
2. Incomplete benchmarking, which omits comparisons to single-cell foundation models (e.g., scGPT) adapted for bulk-data tasks.
3. An absence of critical ablation studies to validate the efficacy of the hybrid CNN-Transformer architecture or the necessity of Gene2Vec initialization.
4. Severe information loss in methylation data processing, where averaging ~450k sites to gene-level features discards crucial, region-specific regulatory patterns.
5. A lack of external validation, as the model's performance is demonstrated only within the TCGA cohort, limiting its proven generalizability.

**Questions:**

N/A

---

> ### Author Response · Authors · 2025-11-26
> **Response to reviewer Yhbd (1)**
>
> Dear Reviewer,
>
> We thank you once again for your review. In addition to the general response addressed to all reviewers to consolidate common feedback, we provide below detailed responses to your specific comments:
>
> 1. >An oversimplified architectural diagram (Fig. 1) that abstracts the core Transformer structure.
>
> We appreciate this feedback. While Figure 1 was intended to provide a high-level overview of the MOJO pipeline—specifically highlighting the U-Net architecture—we acknowledge that it abstracts away the details of the core Transformer structure. To address this, we have added a detailed diagram focused on the Transformer architecture in Section 1 of the supplementary material.
>
>
> 2. >Incomplete benchmarking, which omits comparisons to single-cell foundation models (e.g., scGPT) adapted for bulk-data tasks.
>
> We thank the reviewer for the helpful suggestion. Single-cell foundation models are indeed very popular in the community and are worth considering in the context of this work. However, there are two distinct aspects to consider in this comparison: (i) the data used for pretraining (bulk vs. single-cell RNA-seq) and (ii) the model architecture.
> Regarding architecture, any neural architecture designed for transcriptomic data should, in principle, be applicable to both bulk and single-cell RNA-seq, since the underlying features are shared and the main differences arise from data acquisition. While we did not evaluate our architectures on single-cell data, they are not inherently restricted to bulk RNA-seq. Investigating their performance in the single-cell setting would be interesting future work but is beyond the scope of this study.
> The more central issue is the pretraining distribution. Even with a suitable architecture, a model pretrained entirely on single-cell RNA-seq is unlikely to transfer well to bulk RNA-seq because the two modalities follow very different data distributions (sparsity, noise characteristics, dynamic range, and cell-type mixing). For downstream tasks defined on bulk data, models pretrained on bulk are therefore expected to perform better.
> Still, the reviewer’s comment is valuable, and we agree that testing a representative single-cell foundation model is informative. We therefore fine-tuned scGPT on all downstream tasks. Following prior work on adapting single-cell models to bulk, we attached task-specific heads (classification or survival) and performed parameter-efficient LoRA-like fine-tuning, using exactly the same data splits and hyperparameters as for all other baselines.
>
> *Table 15: Pan-cancer survival analysis with scGPT*
> | Model | Modality | C-index | Weighted C-index |
> |-------|---------|---------|------------------|
> | BulkRNABert | RNA-seq | 0.749 ± 0.003 | 0.654 ± 0.014 |
> | scGPT | RNA-seq | 0.720 ± 0.005 | 0.604 ± 0.020 |
> | MOJO | Bimodal | **0.771 ± 0.006** | **0.670 ± 0.009** |
>
>
> *Table 16: Full benchmark on cancer-type classification with scGPT*
> | Model | Modality | test macro-F1 | test weighted-F1 |
> |-------|----------|---------------|------------------|
> | BulkRNABert | RNA-seq | 0.918 ± 0.008 | 0.943 ± 0.004 |
> | scGPT | RNA-seq | 0.793 ± 0.009 | 0.831 ± 0.008|
> | MOJO | Bimodal | **0.935 ± 0.007** | **0.952 ± 0.006** |
>
>
> Across both tasks, scGPT performs substantially below BulkRNABert and MOJO. We attribute this primarily to the pretraining mismatch: scGPT is pretrained exclusively on single-cell RNA-seq, whereas our downstream tasks are defined on bulk RNA-seq. Earlier in the rebuttal, we also report a full transformer baseline trained from scratch on bulk RNA-seq, effectively equivalent to pretraining a scGPT-style architecture directly on bulk. This baseline also underperforms MOJO, indicating that our architectural choices contribute meaningfully to performance beyond pretraining data alone.
> Overall, we appreciate the reviewer’s suggestion. Including scGPT strengthens our benchmark and supports two conclusions: (i) bulk-specific pretraining is crucial for bulk downstream tasks, and (ii) MOJO’s architecture provides further improvements beyond the choice of pretraining corpus.
>
> References:
>
> [1] Haokun Liu, Derek Tam, Mohammed Muqeeth, Jay Mohta, Tenghao Huang, Mohit Bansal, and Colin A Raffel. Few-shot parameter-efficient fine-tuning is better and cheaper than in-context learning. Advances in Neural Information Processing Systems, 35:1950–1965, 2022
>
> [2] Cui, Haotian, Chloe Wang, Hassaan Maan, Kuan Pang, Fengning Luo, Nan Duan, and Bo Wang. "scGPT: toward building a foundation model for single-cell multi-omics using generative AI." Nature methods 21, no. 8 (2024): 1470-1480
>
> [3] Wang, Hongxiao, Yang Yang, Zhuo Zhao, Pengfei Gu, Nishchal Sapkota, and Danny Z. Chen. "Path-GPTOmic: A balanced multi-modal learning framework for survival outcome prediction." In 2024 IEEE International Symposium on Biomedical Imaging (ISBI), pp. 1-5. IEEE, 2024.

---

> > ### Author Response · Authors · 2025-11-26
> > **Response to reviewer Yhbd (2)**
> >
> > 3. >An absence of critical ablation studies to validate the efficacy of the hybrid CNN-Transformer architecture or the necessity of Gene2Vec initialization.
> >
> > We thank the reviewer for the opportunity to discuss the architectural choices behind MOJO. In response to this and similar feedback from other reviewers, we have included a comprehensive ablation study in the **"MOJO architecture benchmark"** section of our general response. As detailed therein, our experiments confirm that the chosen hybrid architecture, specifically the incorporation of CNNs, achieves superior performance compared to pure Transformer variants while remaining highly computationally efficient.
> >
> >
> > 4. >Severe information loss in methylation data processing, where averaging ~450k sites to gene-level features discards crucial, region-specific regulatory patterns.
> > A lack of external validation, as the model's performance is demonstrated only within the TCGA cohort, limiting its proven generalizability.
> >
> > We thank the reviewer for raising these critical points. As the concerns regarding methylation data processing and external validation were shared by other reviewers, we have addressed them comprehensively in our general response. Please refer to the "Methylation 450k data preprocessing" section for the detailed discussion on information loss, and the "External validation on an independent cohort" section for the generalization analysis. In summary, our additional experiments demonstrate that our efficient methylation averaging strategy remains competitive with more complex region-aware aggregation for downstream tasks, and our external validation on the TARGET dataset successfully confirms the model's robustness and generalization capabilities beyond the TCGA cohort.

---

### Official Review · Reviewer_vC2t · 2025-11-04

**Soundness:** 3
**Presentation:** 3
**Contribution:** 2
**Rating:** 2
**Confidence:** 3

**Summary:**

This paper presents MOJO, a bimodal masked language model that jointly learns from RNA-seq and DNA methylation data using a hybrid CNN–Transformer architecture. It achieves strong performance in cancer classification and survival prediction, remains robust with missing modalities through mutual information loss, and demonstrates biological interpretability consistent with known cancer pathways.

**Strengths:**

The strength of this paper lies in its ability to maintain stable performance even with single-modality or missing-modality data through extended pretraining and mutual information loss. It also demonstrates biological plausibility by combining SHAP and GSEA analyses to verify that the model’s outputs align with known cancer-related pathways.

**Weaknesses:**

1.  Masked language modeling is the standard approach for pretraining in the biomedical domain; it has become a conventional method. The incremental improvement compared to existing approaches (such as BulkRNABert) mainly lies in the bimodal integration, but the technical innovation is limited. Moreover, the accuracy in predicting masked genes may not necessarily reflect biologically meaningful interactions. Is there any related literature to support this?
2. The model only performs a simple concatenation of RNA-seq embeddings and methylation embeddings. I suggest analyzing the similarity between the two modalities, for example, by using KL divergence.

3.  Table 9 shows that even with the addition of the MI loss, the F1 score when RNA-seq data is missing (0.916) is still lower than that of MethFormer (0.931).

4. There are already several existing studies that use multimodal approaches for survival analysis, such as DeepSurv, DeepHit, and RSF. This paper does not include comparisons with them, nor with other works like $\textit{Cross-Modal Translation and Alignment for Survival Analysis}$ (CVPR 2020) and $\textit{Long-term Cancer Survival Prediction Using Multimodal Deep Learning}$ (Scientific Reports, 2021).
5. In the binning strategy, how are the numbers of bins for RNA-seq and methylation  $\(B_{\text{rna}}$, $B_{\text{meth}}\)$ determined?

**Questions:**

No

---

> ### Author Response · Authors · 2025-11-26
> **Response to reviewer vC2t (1)**
>
> Dear Reviewer,
> Thank you for your review. Below, we respond to your specific comments in addition to the general response provided to all reviewers:
>
>
> 1. >Masked language modeling is the standard approach for pretraining in the biomedical domain; it has become a conventional method. The incremental improvement compared to existing approaches (such as BulkRNABert) mainly lies in the bimodal integration, but the technical innovation is limited. Moreover, the accuracy in predicting masked genes may not necessarily reflect biologically meaningful interactions. Is there any related literature to support this?
>
>
> We thank the reviewer for the comment. We do not claim novelty for Masked Language Modeling (MLM); it is a standard objective, and our manuscript states this clearly. The contribution of our work lies instead in the architecture and the joint integration of transcriptomics and DNA methylation, a combination that remains largely unexplored in foundation models. In this multimodal setting, MLM provides a consistent training signal, and we observe distinct learning dynamics across modalities (Appendix A.2, Fig. 6), underscoring the challenge and value of cross-modal pretraining.
> Regarding biological relevance: while MLM does not aim to model causality, several studies show that models trained with this objective learn biologically meaningful structure (West et al., 2023; Theodoris et al., 2023; Karollus et al., 2024). In line with this, we also tested adding a biologically informed prior in the methylation representation and found no improvement, suggesting that the model already captures relevant structure directly from data.
> Most importantly, the learned representations translate into substantial improvements on clinically meaningful downstream tasks (survival prediction, cancer classification), which provides strong evidence of biological and practical relevance.
>
>
> [1] West P, et al. "Genomic language model predicts protein co-regulation and function." Nat Commun. 2023;14:2880.
>
> [2] Karollus A, et al. "Species-aware DNA language models capture regulatory elements and their evolution." Genome Biol. 2024;25:112.
>
> [3] Theodoris CV, et al. "Transfer learning enables predictions in network biology." Nature. 2023;618:616-624.
>
>
> 2. >The model only performs a simple concatenation of RNA-seq embeddings and methylation embeddings. I suggest analyzing the similarity between the two modalities, for example, by using KL divergence.
>
> We thank the reviewer for this insightful suggestion. We acknowledge that RNA-seq and DNA methylation are biologically correlated, given that promoter methylation regulates gene expression, which could theoretically lead to redundancy. To explicitly quantify whether these modalities provide complementary versus redundant information, we computed the Jensen-Shannon (JS) divergence between their value distributions using the raw data, in addition to analyzing their correlation patterns. The results are summarized below:
>
> | Metric | Value |
> |--------|-------|
> | JS divergence per sample (mean ± std) | 0.661 ± 0.011 |
> | JS divergence per gene (mean ± std) | 0.614 ± 0.026 |
> | Genes with JS > 0.6 (distinct distributions) | 67.8% |
> | Pearson correlation (mean ± std) | -0.198 ± 0.156 |
> | Genes with negative correlation | 68.2% |
> | Genes with \|ρ\| > 0.5 | 8.4% |
>
>
> While the modalities are indeed correlated (68% of genes show negative correlation, consistent with established methylation-mediated regulation [1]), the high JS divergence (0.66) demonstrates that they possess fundamentally different value distributions and thus capture complementary information. This confirms that DNA methylation and bulk RNA-seq provide unique, albeit related, views of gene regulation, justifying their combination for multi-modal integration.
>
> [1] Jones PA, Baylin SB. "The fundamental role of epigenetic events in cancer." Nat Rev Genet. 2002;3:415-428.

---

> > ### Author Response · Authors · 2025-11-26
> > **Response to reviewer vC2t (2)**
> >
> > 3. >Table 9 shows that even with the addition of the MI loss, the F1 score when RNA-seq data is missing (0.916) is still lower than that of MethFormer (0.931).
> >
> > We thank the reviewer for this important observation. We acknowledge that even with the inclusion of the mutual information (MI) loss, our bimodal model achieves an F1 score of 0.916 when RNA-seq data is entirely absent, which remains lower than the unimodal MethFormer's 0.931. This gap indicates that specialized unimodal models may retain an advantage when trained and evaluated on a single specific modality. However, we emphasize several key contributions of our proposed approach:
> >
> > Significant performance recovery: while not fully matching the unimodal baseline, the MI loss facilitates substantial performance recovery compared to the bimodal model without MI, demonstrating that our method effectively leverages cross-modal information even in the absence of one modality.
> >
> > Novel fine-tuning paradigm: our application of the mutual information approach operates during the fine-tuning phase, rather than as a test-time adaptation method as seen in [1]. This design ensures that the model learns robust cross-modal representations during training, thereby eliminating the need for computationally expensive test-time optimization and offering a distinct methodological contribution.
> >
> >
> > Scalability: as the number of modalities ($k$) increases, maintaining separate unimodal models for every possible data partition becomes computationally prohibitive (requiring $2^k - 1$ models). Our unified approach offers a scalable solution to this combinatorial challenge.
> >
> > Practical trade-off: the marginal performance difference (0.015 in F1 score) represents a reasonable trade-off for the flexibility of a single unified model capable of handling arbitrary missing modality patterns, particularly as the field scales toward multi-omics data with more than two modalities.
> >
> > We believe these contributions offer significant value to the community, particularly for future multi-modal applications. In such contexts, the combinatorial explosion of missing data patterns renders our unified approach a far more scalable and practical alternative to training separate models for each specific scenario.
> >
> > Reference:
> >
> > [1] Merey Ramazanova, Alejandro Pardo, Bernard Ghanem, and Motasem Alfarra. Test-time adaptation for combating missing modalities in egocentric videos. In The Thirteenth International
> > Conference on Learning Representations, 2025
> >
> >
> > 4. >There are already several existing studies that use multimodal approaches for survival analysis, such as DeepSurv, DeepHit, and RSF. This paper does not include comparisons with them, nor with other works like Cross-Modal Translation and Alignment for Survival Analysis (CVPR 2020) and  Long-term cancer survival prediction using multimodal deep learning (Scientific Reports, 2021).
> >
> > We appreciate the opportunity to clarify our survival analysis methodology and its relationship to the existing literature. Regarding DeepSurv [1], we wish to highlight that our approach builds directly upon this framework; specifically, we employ the DeepSurv architecture and loss function to train our survival head. Since DeepSurv itself represents a neural implementation of the Cox proportional hazards model rather than a specific multimodal integration strategy, our work effectively serves as a bimodal extension of the DeepSurv paradigm. With respect to the additional references cited (Cross-Modal Translation and Alignment, CVPR 2020; and MultiSurv, Scientific Reports 2021), we acknowledge their contributions to multimodal survival analysis. However, both primarily focus on integrating whole slide images (WSI), a modality requiring tailored handling due to distinct dimensionality constraints, using integration techniques comparable to the “Late Integration” baseline already present in our manuscript. To address the comparison with MultiSurv specifically, we implemented its max-pooling integration strategy, benchmarked it against our existing “Late Integration” methods, and have incorporated these results into the updated manuscript.
> >
> > Reference:
> >
> > [1] Jared L Katzman, Uri Shaham, Alexander Cloninger, Jonathan Bates, Tingting Jiang, and YuvalKluger. Deepsurv: personalized treatment recommender system using a cox proportional hazards deep neural network. BMC medical research methodology, 18:1–12, 2018

---

> > > ### Author Response · Authors · 2025-11-26
> > > **Response to reviewer vC2t (3)**
> > >
> > > 5. >In the binning strategy, how are the numbers of bins for RNA-seq and methylation (Brna, Bmeth) determined?
> > >
> > > We thank the reviewer for this important question regarding the binning strategy hyperparameters. We agree that determining the number of bins for RNA-seq ($B_{rna}$) and methylation ($B_{meth}$) is a critical design choice. We provide a detailed explanation and a corresponding ablation study in paragraph 4 of the **"MOJO architecture benchmark"** section of our general response. As detailed therein, our empirical results confirm that our initial selection of bin counts yields superior performance compared to the alternative strategy tested.

---

### Author Response · Authors · 2025-11-26
**General response to reviewers (1)**

Dear Reviewers,
Thank you for your time and thoughtful feedback. We appreciate the clarity of the concerns raised and have structured this general response to address the themes that appeared across multiple reviews. We hope this consolidated overview provides useful context and helps streamline the individual replies.
In response to the comments, we have updated the manuscript and conducted the additional analyses and experiments requested, including new preprocessing steps, pre-training runs, and fine-tuning benchmarks.


### 1. MOJO architecture benchmark

We thank the reviewers for their interest in the efficient MOJO architecture and for suggesting a rigorous benchmark of architectural choices. We specifically address four components:

#### 1.1 Comparison with a pure transformer architecture

Several reviewers asked whether the hybrid design of MOJO, combining a U-Net backbone with transformer layers, provides meaningful advantages over a pure transformer model. To address this, we conducted a full pre-training of a transformer-only variant (“MOJO – Transformer”) using the same data and training setup as MOJO. After pre-training, we fine-tuned this transformer-only model on classification and survival downstream tasks to enable comparisons. Tables 1 and 2 below summarize performance on cancer-type classification and survival analysis, respectively.


*Table 1: MOJO architecture benchmark on cancer-type classification*
| Model | test macro-F1 | test weighted-F1 |
|-------|---------------|------------------|
| MOJO  (no pre-training) | 0.835 ± 0.015 | 0.891 ± 0.006 |
| MOJO  (probing) | 0.928 ± 0.009 | 0.945 ± 0.006 |
| MOJO  | **0.935 ± 0.007** | **0.952 ± 0.006** |
| | | |
| MOJO - Transformer (no pre-training) | 0.802 ± 0.008 | 0.827 ± 0.010 |
| MOJO - Transformer (probing) | 0.850 ± 0.014 | 0.892 ± 0.009 |
| MOJO - Transformer | 0.925 ± 0.005 | 0.942 ± 0.003 |


*Table 2: MOJO architecture benchmark on survival analysis*
| Model | C-index | Weighted C-index |
|-------|---------|------------------|
| MOJO | **0.771 ± 0.006** | **0.670 ± 0.009** |
| MOJO - Transformer | 0.757 ± 0.006 | 0.657 ± 0.007 |

Across all settings (no pre-training, probing, and full fine-tuning), the hybrid MOJO architecture outperforms the pure transformer variant. Beyond accuracy, the computational efficiency gap is substantial: MOJO is over 100× faster in practice (see Table 2 of the manuscript), owing to the downsampling pathway of the U-Net backbone, which dramatically reduces the sequence length processed by the transformer and therefore mitigates the quadratic scaling inherent to attention layers.
Taken together, these results show that the hybrid architecture is not only significantly more efficient, but also more accurate for modeling bulk RNA-seq and DNA methylation data. This provides strong empirical support for the architectural choice. (Further discussion on why multi-scale U-Net processing is effective for unordered gene features is provided below in Section 1.3.)


#### 1.2 Gene2Vec initialization


To address the concern regarding the necessity of Gene2Vec initialization, we re-examined the embedding component of MOJO, which was originally initialized using Gene2Vec embeddings [1]. Following the reviewers’ helpful suggestion, we performed a full pre-training of the model using standard random initialization for all gene embeddings, and subsequently fine-tuned this variant across downstream tasks.


*Table 3: Influence of Gene2Vec initialization on classification performance*
| Model | test macro-F1 | test weighted-F1 |
|-------|---------------|------------------|
| MOJO - with Gene2Vec| 0.935 ± 0.007 | 0.952 ± 0.006 |
| MOJO - no Gene2Vec| 0.932 ± 0.006 | 0.950 ± 0.004 |

*Table 4: Influence of Gene2Vec initialization on survival performance*
| Model  | C-index | Weighted C-index |
|-------|---------|------------------|
| MOJO - with Gene2Vec | 0.771 ± 0.006 | 0.670 ± 0.009 |
| MOJO - no Gene2Vec| 0.771 ± 0.005 | 0.671 ± 0.014 |


Tables 3 and 4 report the results for pancancer classification and survival analysis. The “MOJO – no Gene2Vec” model performs on par with the Gene2Vec-initialized version, with no meaningful differences across evaluation metrics. These findings indicate that Gene2Vec initialization is not required: the model reliably learns effective gene representations directly from the pre-training signal. We thank the reviewers for raising this point, as it helped simplify the architecture. In the revised manuscript, we now recommend standard random initialization.

References:

[1] Quan Zou, Pengwei Xing, Leyi Wei, and Bin Liu. Gene2vec: gene subsequence embedding for prediction of mammalian n6-methyladenosine sites from mrna. Rna, 25(2):205–218, 2019.

---

> ### Author Response · Authors · 2025-11-26
> **General response to reviewer (2)**
>
> #### 1.3 Convolution and absence of natural locality in bulk RNA-seq
>
> We further justify and benchmark our architectural designs, specifically the use of  convolution-based components in a setting where genes do not possess an inherent spatial ordering. We agree that standard convolutions typically assume locality in the input, which is not naturally present in bulk RNA-seq or DNA methylation data. In MOJO, however, we use a U-Net architecture for two specific reasons that are orthogonal to spatial locality.
> First, U-Net provides an efficient hierarchical dimensionality-reduction mechanism, which is essential given the very high input dimensionality (hundreds of thousands of genes or CpGs).
> Second, its multi-scale structure allows extraction of features at different resolutions—from individual gene values in early layers to progressively broader gene aggregates in deeper layers. We believe this multi-scale representation is particularly relevant for transcriptomic data, where both fine-grained and coarse-grained signals carry meaningful information.
> To assess whether explicit biological ordering could further benefit the model, we trained a variant, MOJO-ordered, in which genes were sorted by their genomic coordinates (chromosome number followed by physical position). This imposed a notion of locality that U-Net could, in principle, exploit.
>
> *Table 5: Cancer type classification results - gene ordering benchmark*
> | Model | Modality | Test Macro-F1 | Test Weighted-F1 |
> |-------|----------|---------------|------------------|
> | MOJO | Bimodal | 0.935 ± 0.007 | 0.952 ± 0.005 |
> | MOJO-ordered | Bimodal | 0.936 ± 0.006 | 0.954 ± 0.002 |
>
> *Table 6: Survival analysis results - gene ordering benchmark*
> | Model | Modality | C-index | Weighted C-index |
> |-------|----------|---------|------------------|
> | MOJO | Bimodal | 0.771 ± 0.006 | 0.670 ± 0.009 |
> | MOJO-ordered | Bimodal | 0.775 ± 0.007 | 0.672 ± 0.019 |
>
>
> Results for classification (Table 5) and survival analysis (Table 6) show that MOJO-ordered performs essentially identically to the original model. Differences are small and not statistically significant across tasks. This indicates that the architecture is order-invariant in practice and that its benefits do not rely on genomic adjacency.
> Given these findings, we retain the unordered design. Despite the absence of natural locality, the U-Net structure proves to be an accurate and computationally efficient backbone for high-dimensional omics. Combined with its improved performance relative to alternative models (as shown above in Section 1.1), we view this architecture as a strong and general solution for modeling bulk RNA-seq and DNA methylation data.
>
>
> #### 1.4 Binning strategy for omics tokenization
>
> We thank the reviewers for their question regarding the binning strategy used to tokenize omics data, as this is a key component of MOJO. In our design, bulk RNA-seq and DNA methylation values are discretized into expression levels rather than used as continuous inputs. This choice is motivated by the high noise of raw continuous measurements and the observation that discrete levels are sufficient to capture biologically meaningful patterns. For MOJO, we used 64 bins for both RNA-seq and methylation, matching the configuration used in BulkRNABert and enabling a fair replication of prior results.
> Following the reviewers’ suggestion, we evaluated another binning scheme, specifically whether a substantially lower number of bins could yield comparable performance. Since changing the bin count requires training an entirely new model from scratch, we focused on a representative alternative and pre-trained a “MOJO – 5 bins” variant using the same linear binning scheme. We then fine-tuned this model on all downstream tasks.
>
>
> *Table 7: Influence of the number of bins on classification performance*
> | Model | test macro-F1 | test weighted-F1 |
> |-------|---------------|------------------|
> | MOJO - 64 bins |  **0.935 ± 0.007** | **0.952 ± 0.006** |
> | MOJO - 5 bins  |    0.910 ± 0.007   |     0.937 ± 0.005   |
>
>
> *Table 8: Influence of the number of bins on survival performance*
> | Model | C-index | Weighted C-index |
> |-------|---------|------------------|
> | MOJO - 64 bins | **0.771 ± 0.006** | **0.670 ± 0.009** |
> | MOJO - 5 bins  |   0.758 ± 0.009      |    0.650 ± 0.020   |
>
> As shown in Tables 7 and 8, reducing the number of bins from 64 to 5 consistently decreases performance on both cancer-type classification and survival analysis. While a full sweep across many bin counts would be computationally prohibitive, this comparison provides strong evidence that finer discretization improves model performance. We therefore maintain the 64-bin configuration in the final manuscript.

---

> > ### Author Response · Authors · 2025-11-26
> > **General response to reviewers (3)**
> >
> > ### 2. Methylation 450k data preprocessing
> >
> > To address the concerns regarding potential information loss arising from averaging ~450k CpG sites into gene-level features, we implemented a region-aware weighted aggregation strategy as an alternative to our initial simple mean aggregation. This approach was designed to preserve functionally critical regulatory signals while maintaining compatibility with gene-level RNA-seq features. We assigned differential weights to CpG sites based on their genomic location, reflecting the empirically observed distance-dependent impact of methylation on gene expression. Specifically, promoter-proximal regions (TSS200, TSS1500) received the highest weights (3.0, 2.5, respectively) due to their strong negative correlation with gene expression and established role in transcriptional silencing [1, 2]. 5'UTR and first exon sites were assigned intermediate weights (2.0) based on their impact on transcriptional silencing and elongation inhibition [3], while gene body and 3'UTR regions served as the baseline (weight = 1.0) given their weaker and often positive correlation with transcription [4]. This creates a biologically grounded weighting scheme that prioritizes functionally critical regulatory sites. Additionally, we incorporated CpG island context weights to capture the differential regulatory potential based on proximity to CpG islands. CpG islands (weight = 2.0) were prioritized due to their strong association with gene regulation and promoter activity [5], while CpG shores (weight = 1.5) were weighted to reflect their tissue-specific differential methylation and functional consequences [6]. CpG shelves (weight = 1.0) and open sea regions (weight = 0.8) received lower weights, reflecting their more variable and less consistent associations with gene expression. The final weight for each CpG site $s$ was computed as $w(s) = w_{region}(s) \times w_{island}(s)$, where $w_{region}(s)$ is the genomic region weight and $w_{island}(s)$ is the CpG island context weight, as detailed in the table below.
> >
> > *Table 9: Weights assigned to CpG sites based on genomic region and CpG island context for region-aware methylation aggregation.*
> > | Genomic Region | \(w_{region}\) | CpG Island Context | \(w_{island}\) |
> > |----------------|----------------|-------------------|----------------|
> > | TSS200 | 3.0 | Island | 2.0 |
> > | TSS1500 | 2.5 | Shore | 1.5 |
> > | 5'UTR | 2.0 | Shelf | 1.0 |
> > | 1st Exon | 2.0 | Open Sea | 0.8 |
> > | Gene Body | 1.0 | | |
> > | 3'UTR | 1.0 | | |
> >
> > The weighted methylation value for each gene was then computed as: $X_{meth}[g] = \frac{\sum_{s \in sites(g)} w(s) \cdot X_{sites\_meth}[s]}{\sum_{s \in sites(g)} w(s)}$
> >
> > We trained a full MOJO model using this region-aware aggregation and evaluated it under the same settings as the original model. The downstream performance for cancer-type classification and survival analysis is reported below.
> >
> > *Table 10: Ablation study on weighted average of methylation beta values for cancer type classification*
> > | Non-weighted average of methylation beta value | | Weighted average of methylation beta value | |
> > |---|---|---|---|
> > | **Model** | **Test Weighted-F1** | **Model** | **Test Weighted-F1** |
> > | BulkRNABert* | 0.943 ± 0.004 | BulkRNABert* | − |
> > | scGPT* | 0.831 ± 0.008 | scGPT* | − |
> > | MethFormer | 0.931 ± 0.006 | MethFormer | 0.935 ± 0.005 |
> > | MFA | 0.848 ± 0.008 | MFA | 0.850 ± 0.009 |
> > | NMF | 0.827 ± 0.006 | NMF | 0.824 ± 0.006 |
> > | MOFA | 0.852 ± 0.007 | MOFA | 0.849 ± 0.008 |
> > | IntegrAO | 0.911 ± 0.015 | IntegrAO | 0.944 ± 0.015 |
> > | OmiEmbed | 0.922 ± 0.016 | OmiEmbed | 0.925 ± 0.017 |
> > | Late integration (concat.) | 0.945 ± 0.007 | Late int. (concat.) | 0.940 ± 0.005 |
> > | Late integration (cross-att.) | 0.945 ± 0.002 | Late int. (cross-att.) | 0.941 ± 0.004 |
> > | MultiSurv | 0.911 ± 0.005 | MultiSurv | 0.930 ± 0.010 |
> > | CustOmics (probing) | 0.911 ± 0.088 | CustOmics (probing) | 0.893 ± 0.071 |
> > | CustOmics (e2e**) | 0.946 ± 0.006 | CustOmics (e2e**) | 0.941 ± 0.003 |
> > | MOJO-Transformer (probing) | 0.892 ± 0.009 | MOJO-Transformer (probing) | 0.896 ± 0.007 |
> > | MOJO-Transformer | 0.942 ± 0.003 | MOJO-Transformer | 0.938 ± 0.010 |
> > | MOJO (probing) | 0.945 ± 0.006 | MOJO (probing) |  0.950 ± 0.006 |
> > | **MOJO** | **0.952 ± 0.006** | **MOJO** | **0.951 ± 0.005** |
> > * BulkRNABert and scGPT use only RNA data and are unaffected by methylation preprocessing. \*\* e2e = end-to-end

---

> > > ### Author Response · Authors · 2025-11-26
> > > **General response to reviewers (4)**
> > >
> > > *Table 11: Ablation study on weighted average of methylation beta values for survival analysis*
> > > | Non-weighted average of methylation beta value | | Weighted average of methylation beta value | |
> > > |---|---|---|---|
> > > | **Model** | **Weighted C-index** | **Model** | **Weighted C-index** |
> > > | BulkRNABert* | 0.657 ± 0.011 | BulkRNABert* | − |
> > > | scGPT* | 0.604 ± 0.020 | scGPT* | − |
> > > | MethFormer | 0.618 ± 0.017 | MethFormer | 0.622 ± 0.018 |
> > > | MFA | 0.593 ± 0.016 | MFA | 0.594 ± 0.018 |
> > > | NMF | 0.591 ± 0.025 | NMF | 0.590 ± 0.026 |
> > > | MOFA | 0.601 ± 0.022 | MOFA | 0.598 ± 0.024 |
> > > | IntegrAO | 0.624 ± 0.006 | IntegrAO | 0.625 ± 0.006 |
> > > | OmiEmbed | 0.631 ± 0.007 | OmiEmbed | 0.633 ± 0.007 |
> > > | CustOmics | 0.639 ± 0.099 | CustOmics | 0.642 ± 0.023 |
> > > | Late integration (concat.) | 0.653 ± 0.011 | Late integration (concat.) | 0.652 ± 0.016 |
> > > | MultiSurv | 0.636 ± 0.011 | MultiSurv | 0.635 ± 0.021 |
> > > | MOJO-Transformer | 0.657 ± 0.007 | MOJO-Transformer | 0.641 ± 0.010 |
> > > | **MOJO** | **0.670 ± 0.009** | **MOJO** | **0.654 ± 0.018** |
> > > \* BulkRNABert and scGPT use only RNA data and are unaffected by methylation preprocessing.
> > >
> > >
> > > While the region-aware weighted methylation aggregation strategy was designed to better capture the differential regulatory potential of CpG sites, our empirical evaluation reveals nuanced results. As shown in Tables 9 and 10, this biologically motivated weighting scheme does not yield improvements for the MOJO model in either cancer-type classification or survival prediction tasks; therefore, we retain the simpler average-based aggregation in our final model. When examining the MethFormer baseline, however, we observe marginal improvements with region-aware aggregation. Furthermore, during pre-training, the region-aware scheme achieves consistently higher reconstruction accuracy compared to simple averaging (see Figure 12 in the supplementary material, added to the updated manuscript). This improved reconstruction performance suggests that weighted aggregation, by accounting for the differential regulatory roles of promoters, gene bodies, and CpG islands, yields a more biologically grounded representation of methylation patterns. Nevertheless, these improvements do not transfer to the multimodal setting, indicating that the additional biological signal captured by region-aware weighting may be redundant with information already encoded within the RNA-seq modality.
> > > Given that (1) region-aware aggregation requires significant additional preprocessing, (2) a full pre-training is computationally expensive, and (3) the biologically informed version does not improve the performance of MOJO, we retain the simpler uniform averaging strategy in the final model. This confirms that the model is able to learn the relevant regulatory structure directly from the data without requiring hand-crafted methylation priors.
> > >
> > >
> > >
> > > References:
> > >
> > > [1] Weber M, Hellmann I, Stadler MB, et al. Distribution, silencing potential and evolutionary impact of promoter DNA methylation in the human genome. *Nature Genetics*. 200
> > >
> > > [2] Jones PA, Baylin SB. The epigenomics of cancer. *Cell*. 2007;128(4):683-692.
> > >
> > > [3] Brenet F, Moh M, Funk P, et al. DNA methylation of the first exon is tightly linked to transcriptional silencing. *PLoS ONE*. 2011;6(1):e14524.
> > >
> > > [4] Ball MP, Li JB, Gao Y, et al. Targeted and genome-scale strategies reveal gene-body methylation signatures in human cells. *Nature Biotechnology*. 2009;27(4):361-368.
> > >
> > > [5] Aimée M Deaton and Adrian Bird. Cpg islands and the regulation of transcription. Genes & Development, 25(10):1010–1022, 2011
> > >
> > > [6] Rafael A Irizarry, Christine Ladd-Acosta, Bo Wen, Zhijin Wu, Carolina Montano, Patrick Onyango, Hengmi Cui, Kevin Gabo, Michael Rongione, Maree Webster, et al. The human colon cancer methylome shows similar hypo-and hypermethylation at conserved tissue-specific cpg island shores. Nature Genetics, 41(2):178–186, 2009.

---

> ### Author Response · Authors · 2025-11-26
> **General response to reviewers (5)**
>
> ### 3. External validation on an independent cohort
>
>
> We thank the reviewers for highlighting the importance of evaluating MOJO on data outside of TCGA. In response, we conducted an external validation study using the TARGET dataset, entirely held out from pre-training and fine-tuning, to rigorously assess cross-cohort generalization and exclude any possibility of data leakage. TARGET provides bulk RNA-seq, Illumina450k methylation, and survival data across four pediatric cancer cohorts, summarized in Table 12.
>
> *Table 12: Composition of the TARGET dataset*
> | Cohort | Cancer type | N Patients | Events (deaths) | Censored | Event rate |
> |--------|-------------|------------|-----------------|----------|------------|
> | TARGET-CCSK | Clear Cell Sarcoma of the Kidney | 11 | 3 | 8 | 27.3% |
> | TARGET-NBL | Neuroblastoma | 133 | 66 | 67 | 49.6% |
> | TARGET-OS | Osteosarcoma | 83 | 29 | 54 | 34.9% |
> | TARGET-WT | Wilms Tumor (Kidney) | 122 | 50 | 72 | 41.0% |
> | **Total** | **All Cohorts** | **349** | **148** | **201** | **42.4%** |
>
>
> We benchmarked MOJO and several baselines (BulkRNABert, MethFormer, Late Integration) on survival prediction, zero-shot cancer-type classification, and clustering performance. The results show that MOJO matches BulkRNABert on survival prediction while substantially outperforming late-integration strategies on all zero-shot and clustering metrics, demonstrating strong generalization to a distribution shift.
>
> *Table 13: Survival performance on TARGET dataset*
> | Model | C-index | Weighted C-index |
> |---|---|---|
> | BulkRNABert | **0.601 ± 0.042** | 0.599 ± 0.039 |
> | MethFormer | 0.548 ± 0.027 | 0.540 ± 0.042 |
> | MOJO | **0.601 ± 0.023** | **0.606 ± 0.042** |
>
>
> *Table 14: Zero-shot classification and clustering performance on TARGET dataset*
> | Task | Metric | MOJO | Late integration (concatenation) |
> |------|--------|------|------------------|
> | PAM50 | Acc. | **0.777** | 0.763 |
> |       | NMI  | **0.345** | 0.291 |
> |       | ARI  | **0.213** | 0.154 |
> | Pan-cancer | Acc. | **0.928** | 0.870 |
> |            | NMI  | **0.862** | 0.771 |
> |            | ARI  | **0.756** | 0.620 |
> | TARGET | Acc. | **0.992** | 0.991 |
> |        | NMI  | **0.949** | 0.794 |
> |        | ARI  | **0.946** | 0.706 |
>
> To further evaluate cross-platform robustness, we quantified batch effects between TCGA and TARGET using standard metrics: (1) Silhouette scores to measure global dataset separation (range: -1 to 1, with 0 indicating complete mixing), (2) the k-nearest neighbor Batch Effect Test (kBET) to assess whether batch labels are randomly distributed in local neighborhoods (range: 0 to 1, with 1 indicating perfect mixing), and (3) the Pearson correlation of mean bulk RNA-seq and DNA methylation levels between platforms. Our analysis revealed weak batch effects (bulk RNA-seq: Silhouette=0.19, kBET=0.95, r=0.86; DNA Methylation: Silhouette=0.12, kBET=0.92, r=0.96). Additionally, PCA visualization showed substantial overlap between TCGA and TARGET samples (Appendix D.3, Figure 13). The high kBET scores and strong correlations demonstrate excellent local mixing and overall concordance, validating the model's cross-platform generalization capabilities.
> Together, this external validation demonstrates that MOJO generalizes robustly to an independent dataset with different demographics, cancer types, and experimental conditions, providing strong evidence of its stability and real-world applicability.

---

> ### Author Response · Authors · 2025-11-26
> **General response to reviewers (6)**
>
> ### 4. Benchmarking against Single-Cell Foundation Models (scGPT)
>
> Reviews suggested comparing MOJO to foundation models originally developed for single-cell RNA-seq. We agree that this is an important baseline to assess, as recent work has shown that single-cell models can sometimes be adapted to bulk data through tailored fine-tuning strategies (Path-GPTOmic [3]).
> Although bulk and single-cell RNA-seq originate from fundamentally different data distributions—and we believe bulk-specific architectures remain necessary—we incorporated scGPT [2] into our benchmarking to provide a complete and fair evaluation. Following prior work on domain adaptation for bulk RNA-seq, we adapted scGPT by attaching task-specific heads (for classification or survival) and applying Lora-like parameter-efficient fine-tuning [1], matching the fine-tuning setup used for all other baselines.
> We then fine-tuned scGPT on downstream tasks using exactly the same data splits and hyperparameter settings as for the other models.
>
>
> *Table 15: Pan-cancer survival analysis with scGPT*
> | Model | Modality | C-index | Weighted C-index |
> |-------|---------|---------|------------------|
> | BulkRNABert | RNA-seq | 0.749 ± 0.003 | 0.654 ± 0.014 |
> | scGPT | RNA-seq | 0.720 ± 0.005 | 0.604 ± 0.020 |
> | MOJO | Bimodal | **0.771 ± 0.006** | **0.670 ± 0.009** |
>
>
> *Table 16: Full benchmark on cancer-type classification with scGPT*
> | Model | Modality | test macro-F1 | test weighted-F1 |
> |-------|----------|---------------|------------------|
> | BulkRNABert | RNA-seq | 0.918 ± 0.008 | 0.943 ± 0.004 |
> | scGPT | RNA-seq | 0.793 ± 0.009 | 0.831 ± 0.008|
> | MOJO | Bimodal | **0.935 ± 0.007** | **0.952 ± 0.006** |
>
> Across both survival prediction and cancer-type classification, scGPT performs substantially below BulkRNABert and MOJO. This confirms that, despite its strength in single-cell representation learning, scGPT does not transfer effectively to bulk RNA-seq without dedicated architectural adaptations. Including this additional baseline strengthens our overall benchmarking and supports the need for bulk-specific models such as MOJO.
>
> [1] Haokun Liu, Derek Tam, Mohammed Muqeeth, Jay Mohta, Tenghao Huang, Mohit Bansal, and Colin A Raffel. Few-shot parameter-efficient fine-tuning is better and cheaper than in-context learning. Advances in Neural Information Processing Systems, 35:1950–1965, 2022
>
> [2] Cui, Haotian, Chloe Wang, Hassaan Maan, Kuan Pang, Fengning Luo, Nan Duan, and Bo Wang. "scGPT: toward building a foundation model for single-cell multi-omics using generative AI." Nature methods 21, no. 8 (2024): 1470-1480
>
> [3] Wang, Hongxiao, Yang Yang, Zhuo Zhao, Pengfei Gu, Nishchal Sapkota, and Danny Z. Chen. "Path-GPTOmic: A balanced multi-modal learning framework for survival outcome prediction." In 2024 IEEE International Symposium on Biomedical Imaging (ISBI), pp. 1-5. IEEE, 2024.
>
> ### 5.Concluding remarks on the general comments
>
> In summary, we have substantially extended the empirical evaluation of MOJO in response to the reviewers’ comments. Beyond the original submission, we (i) pre-trained and fine-tuned a pure transformer variant, a no-Gene2Vec variant, a genomically ordered variant, and a low-bin (5-bin) variant of MOJO on all downstream tasks; (ii) re-ran the full pipeline with an alternative, biologically motivated methylation aggregation scheme; (iii) added a strong new baseline based on the single-cell foundation model scGPT, adapted and fine-tuned for bulk RNA-seq; and (iv) performed external validation on the independent TARGET cohort, including a dedicated analysis of cross-platform batch effects. Together, these experiments provide converging evidence that our architectural choices, binning strategy, and simple methylation aggregation are well justified, and that MOJO achieves robust and generalizable performance across tasks, modalities, and cohorts.
> We are grateful to the reviewers for the constructive feedback that motivated this additional work and helped strengthen the manuscript. We now proceed to address the specific comments of each reviewer individually.

---

### Meta-Review · Area_Chair_U8Cz · 2025-12-09

**Summary:**

This work introduces MOJO, a bimodal masked language model for RNA-seq and DNA methylation data. Four reviewers thoroughly reviewed the submitted paper, with one reject (rating 2), two marginally below the acceptance threshold (rating 4), and one marginally above the acceptance threshold (rating 6). The overall rating was below the acceptance bar, and the reviewers were inclined towards the negative aspects.

Reviewer vC2t raised concerns that the MLM is standard, and the novelty and contributions are not significant. I agree with this perspective. From the authors’ response, they stated that the contribution lies in the application to transcriptomics and DNA methylation data, which is under-explored. I partially accept this response, but the concern of technical advancements is still not well-addressed. The reviewer also highlighted some related works, including DeepSurv, and the authors agreed that their work was built upon it. This suggests that the technical innovation may be incremental. For other concerns, I think the authors addressed well with comprehensive results and discussions.

Reviewer Yhbd was mainly concerned about the validation results, including ablation studies and external evaluation. The authors responded with added experiments in the rebuttal. I think this response is acceptable.

Reviewer GBM6 also raised concerns about novelty and experiments. Similarly, the authors added experiments to reply. However, the authors did not reply to the concerns of novelty. I have to say that this concern is still not resolved.

Reviewer Rh7r first asked the question of data leakage. I think this question is still not well addressed. The authors agree that *“utilizing the TCGA cohort for both pre-training and downstream evaluation constitutes a limitation”*. External validation can only partially solve the problem, while the limitation of internal validation still exists. For other comments like ablation studies and other experimental settings, I think the authors addressed them well.

In summary, while this paper has its strengths and the authors provided extensive experiments in the rebuttal, its key contributions and novelty remain unclear. Given the overall negative feedback from reviewers and the absence of sufficiently compelling merits for acceptance, considering the high bar of a top-tier conference like ICLR, I am sorry to ultimately recommend rejection.

**Reviewer Concerns:**

Reviewers Yhbd and Rh7r's comments are resolved. Reviewers vC2t and GBM6's are still outstanding.

**Reviewer Scores:**

I think all reviewers will keep their scores.

---

### Decision · Program_Chairs · 2026-01-26

Reject